


# Bering Sea surface water conditions during Marine Isotope Stages 12 to 10 at Navarin Canyon (IODP Site U1345)

**Beth E. Caissie[1], Julie Brigham-Grette[2], Mea S. Cook[3], Elena Colmenero-Hidalgo[4]**

[1]{Iowa State University, Ames, Iowa}

[2]{University of Massachusetts, Amherst, Amherst, Massachusetts}

[3]{Williams College, Williamstown, Massachusetts}

[4]{Universidad de León, León, Spain}

Correspondence to: B.E. Caissie (bethc@iastate.edu)

## Abstract

Records of past warm periods are essential for understanding interglacial climate system dynamics. Marine Isotope Stage 11, occurred ~410 ka when global ice volume was the lowest, sea level was the highest and terrestrial temperatures were the warmest of the last 500 kyrs. This interval with its extreme character has been considered an analog for the near future. The Bering Sea is ideally situated to record how opening or closing the Pacific-Arctic Ocean gateway (Bering Strait) impacted primary productivity, sea ice, and sediment transport in the past; however, little is known about this region prior to 125 ka. IODP Expedition 323 to the Bering Sea offered the unparalleled opportunity to look in detail at time periods older than had been previously retrieved using gravity and piston cores. Here we present a multi-proxy record for Marine Isotope Stages 12-10 from Site U1345 located near the shelf-slope break. MIS 11 is bracketed by highly productive laminated intervals that may have been triggered by flooding of the Beringian shelf. Low insolation is associated with higher productivity, which was likely driven by increased upwelling. During the majority of MIS 11 however, high stratification appears to have



led to lowered productivity in both the northern Atlantic and the northern Pacific. U1345,
located near the marginal ice zone, experienced seasonal sea ice throughout both the
glacial and interglacial stages. When global eustatic sea level was at its peak, Beringian
tidewater glaciers advanced, driven by decreasing insolation, reduced seasonality, and
high humidity due to high sea level and ice-free summers. Multiple examples of Pacific-
Atlantic teleconnections are presented including laminations deposited at the end of MIS
11 in sync with millennial scale stadial events seen in the North Atlantic.

## 1    Introduction

Predictions and modeling of future climate change require a detailed understanding of
how the climate system works. Reconstructions of previous warm intervals shed light on
interhemispheric teleconnections. The most recent interglacial period when orbital
conditions were similar to today was approximately 400 ka during the extremely long
interglacial known as marine isotope stage (MIS) 11. Eccentricity was low, obliquity was
high and the amplitude of precessional changes was low (Loutre and Berger, 2003). In
addition, $CO_2$ concentration averaged approximately 275 ppm, which is similar to pre-
industrial levels (EPICA community members, 2004). The transition from MIS 12 into
MIS 11 has been compared to the last deglaciation (Dickson et al., 2009) and extreme
warmth during MIS 11 has been considered an analog for future warmth (Droxler et al.,
2003; Loutre and Berger, 2003), although the natural course of interglacial warmth today
has been disrupted by anthropogenic forcing (IPCC, 2013).
Globally, MIS 11 is easily recognizable in the sediment record by an abrupt and distinct
transition from high to low $\delta^{18}O$ values at the MIS 12/11 boundary and subsequent
prolonged low values of $\delta^{18}O$ during MIS 11 (Lisiecki and Raymo, 2005). Furthermore,
MIS 11 was unique in the polar regions. Antarctica experienced temperatures 2° C
warmer than pre-industrial temperatures (Jouzel et al., 2007), and boreal forest extended
across Greenland, which may have been largely ice free (de Vernal and Hillaire-Marcel,
2008). Large lakes in Siberia were anomalously productive and record warmer air and
lake temperatures than today. Lake Baikal was 2° C warmer (Prokopenko et al., 2010)
and Lake El'gygytgyn was 4° C warmer (Lozhkin and Anderson, 2013; Vogel et al.,



2013). MIS 11 is also unique in Beringia because coastal glaciers advanced midway
through the long interglacial cycle while sea level was still high (Brigham-Grette et al.,
2001; Huston et al., 1990; Kaufman et al., 2001; Pushkar et al., 1999). This implies that
parts of Beringia were glaciated rapidly as high latitude insolation fell in the northern
hemisphere, but before global sea level dropped in response to the buildup of large ice
sheets reaching lower latitudes. MIS 11 ice (i.e., leading to the Nome River Glaciation in
MIS 10) is widely believed to be the last of the most extensive glaciations in central
Beringia (Brigham-Grette et al., 2001; Gualtieri and Brigham-Grette, 2001; Kaufman et
al., 1991; Manley et al., 2001).
Despite the work done to characterize the warmth of MIS 11 in the terrestrial realm
(Candy et al., 2014; Melles et al., 2012; Prokopenko et al., 2010), as well as the North
Atlantic (Bauch et al., 2000; Chaisson et al., 2002; Dickson et al., 2009; Milker et al.,
2013; Poli et al., 2010), little is known about this interval from the Pacific region (Candy
et al., 2014). Modeling studies describe several mechanisms for linking the Atlantic and
Pacific through oceanic heat transport on glacial-interglacial scales (DeBoer and Nof,
2004; Hu et al., 2010), however, there have been no tests of these modeling studies using
proxy data older than 30 ka. Furthermore, the location of the Bering Sea marginal ice
zone advanced and retreated hundreds of kilometers during the past three glacial-
interglacial cycles (Caissie et al., 2010; Katsuki and Takahashi, 2005; Sancetta and
Robinson, 1983); however, sea surface and intermediate water variability before MIS 5 is
unknown.
This investigation of terrestrial-marine coupling at the shelf-slope break from MIS 12 to
10 is the first study of this interval in the subarctic Pacific (Fig. 1). We use
sedimentology, diatom assemblages, geochemistry, and calcareous nannofossil
abundances as proxies for sea ice, sea surface conditions, and shelf to basin transport.
These proxy records show changes in sea ice and glacial ice that are in sync with
insolation changes at high northern latitudes. An interval of productivity occurs at the
glacial termination (Termination V) that is both short-lived and intense. This is followed
by evidence of glacial advance in Beringia during peak MIS 11 when sea level is high.
We put these changes seen in the Bering Sea in the context of global records of MIS 11.






## 2  Background

### 2.1 Global Sea Level during MIS 11

The maximum height of sea level during MIS 11 is an open question with estimates ranging from more than 20 m above present sea level (apsl) (Kindler and Hearty, 2000) to 0 m apsl (Rohling et al., 2010). The discrepancy may stem from large differences between global eustatic (Bowen, 2010) or ice-volume averages (McManus et al., 2003) and regional geomorphological or micropaleontological evidence (van Hengstum et al., 2009). Regional isostatic adjustment due to glacial loading and unloading is also perhaps not as insignificant as previously assumed and regional highstands may record higher than expected sea levels if glacial isostasy and dynamic topography have not been accounted for, even in places that were never glaciated (Raymo and Mitrovica, 2012; Raymo et al., 2011). Work by Raymo and Mitrovica suggests that eustatic sea level during MIS 11 was 6-13 m apsl globally and near 5 m apsl locally in Beringia (Raymo and Mitrovica, 2012).

Regardless of the ultimate height of sea level, the transition from MIS 12 to MIS 11 was the greatest change in sea level of the last 500 ka (Rohling et al., 1998); sea level rose from -140 m to its present level or higher (Bowen, 2010). Up to three highstands occurred during MIS 11 (Kindler and Hearty, 2000), the highest at the second June insolation peak at 65°N. This highstand is referred to as MIS 11.3 and occurs 406 ka (Bassinot et al., 1994). Sea levels above modern require partial or complete collapse of the Greenland ice sheet (up to 6 m) (de Vernal and Hillaire-Marcel, 2008) and/or the West Antarctic Ice Sheet (Scherer et al., 1998), but not the East Antarctic Ice Sheet (Berger et al., 2015; Raymo and Mitrovica, 2012). It has frequently been hypothesized that the West Antarctic Ice Sheet collapsed during MIS 11 and modeling studies confirm this (Pollard and DeConto, 2009), however drilling in Antarctica was not able to confirm a collapse (Naish et al., 2009). East Antarctica appears stable; however, small changes in either Antarctic ice sheet may have contributed up to 5 m of sea level rise (Berger et al., 2015; EPICA community members, 2004).

117



## 2.2 Sea Level Variation in Beringia

Beringia refers to both the terrestrial and marine regions north of the Aleutian Islands that stretch to the shelf-slope break in the East Siberian, Chukchi, and Beaufort seas (Fig. 1). On land, Beringia extends from the Lena River in Siberia to the Mackenzie River in Canada. Large portions of the Beringian shelf are exposed when sea level drops below -50 m (Hopkins, 1959) and this subaerial expanse stretches more than 1000 km from north to south during glacial periods (Fig. 2). In contrast, as sea level rises at glacial terminations the expansive continental shelf is flooded, introducing fresh organic matter and nutrients into the southern Bering Sea (i.e. Bertrand et al., 2000; Ternois et al., 2001) and re-establishing the connection between the Pacific and Atlantic oceans through Bering Strait.

## 2.3 Beringian Hydrography

Today, water circulates cyclonically in the deep basins of the Bering Sea (Fig. 1). Site U1345 is influenced by the northwest flowing Bering Slope Current, which is derived from the Alaskan Stream. South of the Aleutians Islands, the Alaskan Stream flows westward and enters the Bering Sea through deep channels in the western Aleutian Islands. Once north of the Aleutian Islands, this water mass is called the Aleutian North Slope Current, and flows eastward until it reaches the Bering Sea shelf. Interactions with the shelf turn this current to the northwest where it becomes the Bering Slope Current (Stabeno et al., 1999). Tidal forces and eddies in the Bering Slope Current drive upwelling through Navarin Canyon and other interfluves along the shelf-slope break (Kowalik, 1999). The resulting cold water and nutrients brought to the sea surface, coupled with the presence of seasonal sea ice, drive the high productivity found today in the so called "Green Belt" (Springer et al., 1996) along the shelf-slope break. North of the site, low salinity, high nutrient shelf waters (Cooper et al., 1997) primarily flow north through the Bering Strait to the Arctic Basin (Schumacher and Stabeno, 1998).

Major changes in circulation may have occurred throughout MIS 12 to 10 due to sea level rise and fall, the strength of North Atlantic Deep Water (NADW) formation, and the intensity of winds originating in the Southern Ocean (e.g. DeBoer and Nof, 2004; Hu et



al., 2010). Specifically, as sea level rose after MIS 12, the connection between the Pacific
and the Atlantic was reestablished. De Boer and Nof (2004) suggest that under high sea
level conditions, if freshwater is suddenly released into the North Atlantic, the Bering
Strait might act as an "exhaust valve" allowing fresh water from the North Atlantic to
flow into the Arctic Ocean and then flow south through the Bering Strait, thus preventing
a shut-down in thermohaline circulation (DeBoer and Nof, 2004). Hu et al. (2010)
suggest that when sea level is fluctuating near the sill depth of Bering Strait (~ -50 m apsl
today), this gateway can modulate widespread climate changes. When Bering Strait is
open and North Pacific water is transported to the North Atlantic, the less saline Pacific
water can freshen the North Atlantic and slow meridional overturning, subsequently
cooling the North Atlantic, but warming the North Pacific. This North Atlantic cooling
can result in the buildup of ice sheets and global sea level drops, closing Bering Strait. A
closed Bering Strait concentrates fresher waters in the North Pacific and more saline
waters in the North Atlantic. A more saline Atlantic means that stratification is low and
meridional overturning is increased. This increases oceanic heat transport to the North
Atlantic, triggers warming in the North Atlantic and ultimately ice sheets retreat and sea
level rises reconnecting the Pacific with the Atlantic (Hu et al., 2010).

## 3   Methods

### 3.1 Study Area and Sampling

The Integrated Ocean Drilling Program's (IODP) Expedition 323, Site U1345, is located
on an interfluve ridge near the shelf-slope break in the Bering Sea (Fig. 1). Navarin
Canyon, one of the largest submarine canyons in the world (Normack and Carlson, 2003)
is located just to the northwest of the site. Sediments were retrieved from ~1008 m of
water, placing the site within the center of the modern day oxygen minimum zone
(Takahashi et al., 2011). We focus on this site because of its proximity to the modern
marginal ice zone in the Bering Sea and observed high sedimentation rates.
Site U1345 was drilled five times during Exp. 323 and cores from four of these holes
were described onboard the JOIDES Resolution. This study focuses on a splice of 3 holes
that were correlated onboard the ship, so that core gaps in one hole are covered by core



material in other holes. In addition to the original analyses presented here, we refer to the
shipboard core descriptions and physical properties data (Takahashi et al., 2011) in our
interpretations. Depths are reported in CCSF-A, a correlated depth scale that allows for
direct comparison between drill holes. Units are meters below sea floor (mbsf). A small
syringe was used to collect approximately 1 cc of sediment periodically between 112.96
m and 136.40 mbsf. Sampling resolution varied for each analysis. Hyalochaete
*Chaetoceros* resting spores were counted on average every 20 cm (~600 yr resolution),
full diatom counts were carried out every 36 cm (~1000 yr resolution), calcareous
nannofossils were counted every 40 cm (~1200 yr resolution), grain size was analyzed
every 23 cm (670 yr resolution), and geochemistry was analyzed every 30 cm (800 yr
resolution).

**3.2 Age Model**
The age model (Fig. 3) is derived from the shipboard age model, which was developed
using magnetostratigraphy and biostratigraphy. First and last appearance datums for
diatoms and radiolarians make up the majority of the biostratigraphic markers used to
place the record in the correct general stratigraphic position (Takahashi et al., 2011).
Oxygen isotope measurements taken on the benthic foraminifera, *Uvigerina peregrina,*
*Nonionella labradorica, and Globobulimina affinis* (Cook et al., In Press) were then used
to tune site U1345 to the global marine benthic foraminiferal isotope stack (LR04)
(Lisiecki and Raymo, 2005) (Fig. 3). Based on this combined age model, MIS 11 spans
from 115.3 to 130.6 mbsf (Cook et al., In Press); however, the characteristic interglacial
isotopic depletion was not found in U1345 which means that the exact timing of peak
interglacial conditions is unknown.
The nearby core, IODP Exp 323, Site U1343 (Fig. 1) has an excellent oxygen isotopic
record during MIS 11 (Kim et al., 2014). We compared the two isotopic records and their
magnetic susceptibilities (Fig. 3) and found that even with only two tie points, there was
good correlation between the timing of the onset of laminated intervals and also the
interglacial increase in magnetic susceptibility (Fig. 3). We added one additional tie point
to connect the inflection points in magnetic susceptibility. In U1343, this point occurred



at 398.50 ka. U1345 was shifted 1.5 ka younger in order to align with U1343. The
addition of this point allows us to have more confidence in the timing of peak interglacial
conditions in U1345. However, given the oxygen isotope sampling resolution, as well as
the stated error in the LR04 dataset (4 kyr), we estimate the error of the age model could
be up to 5 kyr. Therefore, we urge caution when interpreting millennial scale changes at
the site or comparing our record to others that examine MIS 11 at millennial scale
resolution or finer.
Sedimentation rates during the study interval range from 29 cm/kyrs to 45 cm/kyr with
the highest sedimentation rates occurring during glacial periods.

**3.3 Diatom Analysis**

In order to quantify the number of diatom valves deposited per gram of sediment, diatom
slides were prepared according to the method described in Scherer (1994). This method
allows for a direct comparison of diatom accumulation in the sediments between samples,
though not diatom flux. Cover slips were mounted on cleaned microscope slides using
hyrax in toulene (refractive index: 1.7135). At least 300 diatom valves were identified in
at least three random transects across the slide using a light microscope at magnifications
from 1000x to 1250x (see Armand et al., 2005; Sancetta, 1979; Sancetta and Silvestri,
1986; Scherer, 1994). The portion of the slide that was examined was measured using a
stage micrometer. Partial valves were counted according to the methods of Schrader and
Gersonde (1978). All diatoms were identified to the species level when possible
following published taxonomic descriptions and images (Hasle and Heimdal, 1968;
Koizumi, 1973; Lundholm and Hasle, 2008, 2010; Medlin and Hasle, 1990; Medlin and
Priddle, 1990; Onodera and Takahashi, 2007; Sancetta, 1982, 1987; Syvertsen, 1979;
Tomas, 1996; Witkowski et al., 2000). Diatom counts were transformed into relative
percent abundances. Absolute abundances (diatoms per gram sediment) were calculated
following the methods of Scherer (1994). Diatom taxa were then grouped according to
ecological niche (Table 1) based on biological observations (Aizawa et al., 2005; Fryxell
and Hasle, 1972; Håkansson, 2002; Horner, 1985; Saito and Taniguchi, 1978;
Schandelmeier and Alexander, 1981; von Quillfeldt, 2001; von Quillfeldt et al., 2003)





and statistical associations (Barron et al., 2009; Caissie et al., 2010; Hay et al., 2007; Katsuki and Takahashi, 2005; Lopes et al., 2006; McQuoid and Hobson, 2001; Sancetta, 1982, 1981; Sancetta and Robinson, 1983; Sancetta and Silvestri, 1986; Shiga and Koizumi, 2000). In cases where a diatom species was reported to fit into more than one environmental niche, it was grouped into the niche where it was most commonly recognized in the literature.

### 3.4 Calcareous Nannofossils

In order to quantify calcareous nannofossils per gram of sediment, a total of 18 samples were prepared following the methodology of Flores and Sierro (1997). A known mass of dried sediment was diluted in a known volume of buffered water. A small fraction was extracted with a micropipette and dropped onto a petri dish previously filled with buffered water and with a cover slip in its bottom. After settling overnight, the excess water was removed and the slide was left to dry and then mounted using Canada balsam.

Observations were made using a Zeiss polarized light microscope at 1000x magnification. Samples were considered barren if no coccoliths were found in at least 165 randomly selected fields of view. All taxa were identified to the species or variety level, following Flores et al. (1999) and Young et al. (2003).

### 3.5 Grain Size

Volume percent of grains in 109 size bins ranging from 0.01 µm to 3500 µm was measured using a Malvern Mastersizer 3000 with the Hydro MV automated wet dispersion unit. Samples were prepared by adding 200 µl of the deflocculant, sodium hexametaphosphate, to 0.2 mg dry sediment. In this way, we were able to quantify all sediment types including biogenic and terrigenous grains.



### 3.6 Geochemistry

Sediment samples were freeze-dried then ground. An aliquot of homogenized sediment was treated to remove carbonates using pH 5 buffered acetic acid. The carbon (from the acidified sediment) and nitrogen (from the unacidified sediment) isotopic and elemental composition of organic matter was determined by Dumas combustion using a Carlo Erba 1108 elemental analyzer coupled to a Thermo-Finnigan Delta Plus XP isotope ratio mass spectrometer at the University of California Santa Cruz Stable Isotope Laboratory. The 1-sigma precision of stable isotope measurements and elemental composition of carbon are 0.2‰ and 0.03%, respectively, and for nitrogen are 0.2‰ and 0.002%, respectively. Percent $CaCO_3$ was calculated according to Schubert and Calvert (2001).

## 4 Results

### 4.1 Sedimentology

In general the sediments at Site U1345 are massive with centimeter–scale dark or coarse-grained mottles. The sediments are mainly composed of clay and silt with varying amounts of diatoms, sand, and tephra throughout. Laminated intervals bracket MIS 11 (Fig. 4). The proportion of diatoms relative to terrigenous or volcanogenic grains is highest during laminated intervals and lowest immediately preceding Termination V (~425 ka). Vesiculated tephra shards were seen in every diatom slide analyzed. Several thin (< 1 cm) sand layers and shell fragments were visible on the split cores, especially during MIS 12. However, high-resolution grain size analyses show that the median grain size was lowest during MIS 12, increasing from approximately 14 μm to 21 μm at the start of Termination V at 424.5 ka (130.92 mbsf). Median grain size peaks at 84 μm between 401 and 407 ka (125.42-123.62 mbsf). This interval is also the location of an obvious sandy layer in the core. After this interval, median grain size remains steady at about 17 μm. Subrounded to rounded clasts (granule to pebble) commonly occur on the split surface of the cores. We combined clast and sand layer data from all Holes at Site U1345 when examining their distribution (Fig. 4).



A 3.5 m thick laminated interval, estimated to span 12 kyrs (see Table 2 for depths and
ages) is deposited beginning during Termination V. Although the termination is short
lived and the laminated interval quite long, we refer to it as the Termination V
Laminations for the sake of clarity throughout this manuscript. The next laminated
interval occurs at about 394 ka and lasts approximately 1100 years. During the transition
from late MIS 11 to MIS 10, a series of four thin laminated intervals are observed. Each
interval lasts between 0.34 and 1.25 ka (Table 2). In general, the upper and lower
boundaries of laminated intervals are gradational; however the boundaries between
individual lamina are sharp (Takahashi et al., 2011). There are two types of laminations.
The Termination V Laminations are Type I laminations: millimeter-scale alternations of
black, olive gray, and light brown triplets. In addition to containing a high proportion of
diatoms, this laminated interval also contains high relative proportions of calcareous
nannofossils and foraminifera (Takahashi et al., 2011). The majority of laminations are
parallel; however, a 7 cm interval during the Termination V Laminations is highly
disturbed in Hole A, showing recumbent folds in the laminations (Takahashi et al., 2011).
This interval was not sampled. Type II laminations occur throughout the remainder of
MIS 11. These laminations have fewer diatoms and tend to be couplets of siliciclastic
sediments with <40% diatoms (Takahashi et al., 2011). Percent $CaCO_3$ also increases
during these laminations though foraminifera and calcareous nannofossils are very rarely
seen. None of these later laminated intervals contain any evidence of disturbance.

### 4.2 Diatoms

#### 4.2.1  Diatom Assemblages

A total of 97 different diatom taxa were identified. Individual samples include between
26 and 46 taxa each with an average of 37 taxa. Both types of laminated intervals contain
fewer taxa than bioturbated intervals do. This decrease in diversity is confirmed using the
Margalef, Simpson, and Shannon indices which all show similar down-core profiles (Fig.
5). The Margalef index is a measure of species richness (Maurer and McGill, 2011). It
shows a decrease in the number of taxa during four out of five laminated intervals that are
sufficiently well sampled. Between laminated intervals, there is a noted decrease in taxa





at 388 ka. Instead of species richness, the Simpson index measures the evenness of the
sample. Values close to 1 indicate that all taxa contain an equal number of individuals,
while values close to 0 indicate that one species dominates the assemblage (Maurer and
McGill, 2011). In general, the Simpson index is close to 1 throughout the core indicating
a rather even distribution of diatom valves across all taxa; however, during the
Termination V Laminations and the most recent two laminations, the Simpson index
drops reflecting the dominance by *Chaetoceros* RS during these intervals (Fig. 5). It does
not come close to 0, which would likely indicate a strong dissolution signal. The Shannon
diversity index measures both species richness and evenness (Maurer and McGill, 2011).
Correspondingly, it is low during three of the laminated intervals, high during MIS 12
and peaks at 397 ka (Fig. 5).
Absolute diatom abundances vary between $10^6$ and $10^8$ diatoms deposited per gram of
sediment with values an order of magnitude higher during most laminated intervals than
during massive intervals (Fig. 5). The diatom assemblage is dominated by *Chaetoceros*
and *Thalassiosira antarctica* resting spores (RS), with lesser contributions from
*Fragilariopsis oceanica, Fragilariopsis cylindrus, Fossula arctica, Shionodiscus trifultus*
(=*Thalassiosira trifulta), Thalassiosira binata,* small (<10 μm in diameter) *Thalassiosira*
species, *Paralia sulcata, Lindavia* cf. *ocellata, Neodenticula seminae,* and *Thalassionema*
*nitzschioides,* (Fig. 6).
Relative percent abundances of *Chaetoceros* RS are highest (up to 69%) during the
Termination V Laminations and, in general, mimic the pattern of both diatom
accumulation rate and insolation at 65° N (Berger and Loutre, 1991). When insolation is
low, *Chaetoceros* RS are also low (Fig. 7). *T. antarctica* RS, in contrast, are lowest
during the Termination V Laminations (as low at 1%) and higher during MIS 12 and after
406 ka (above 125.00 mbsf and 112.97 mbsf). This taxon peaks at 38% relative
abundance at 390 ka (120.45 mbsf; Fig. 6).
Relative percent abundances of the characteristic marginal ice zone species, *F. oceanica*
and *F. cylindrus* (Caissie et al., 2010; Saito and Taniguchi, 1978; Sancetta, 1982; von
Quillfeldt et al., 2003), oscillate between ~10% and less than 3% of the diatom
assemblage and are highest during MIS 12 and all laminated intervals. They are both at



their lowest between ~411 to ~400 ka (126.62 to 123.45 mbsf). The neritic species and
moving water indicator, *P. sulcata* is lowest during the laminated intervals. It reaches a
maximum (34% relative abundance) at 404 ka (124.61 mbsf). *P. sulcata* remains
moderately high (~10%) during non-laminated intervals. *L.* cf. *ocellata* is the dominant
taxa in the fresh water group and the variability in its abundances is discussed below. *S.*
*trifultus* follows a very similar distribution to the fresh water group and *L.* cf. *ocellata*. It
is relatively high (~4%) during MIS 12, is virtually absent from the sediments during the
Termination V Laminations, and then increases again until it peaks at 10% relative
abundance at 400 ka (123.22 mbsf). *Thalassiosira binata* and other small (<10 μm in
diameter) *Thalassiosira* species have similar distributions with low relative abundances
throughout the record (< 6%) except for a small peak between 397 and 386 ka (122.62
and 119.07 mbsf. The relative percent abundances of *N. seminae* are discussed below.
The largest peak in *N. seminae* is at 392 ka (121.2 mbsf) (Fig. 6).

### 366  4.2.2 Diatom Proxies

Diatoms, like many organisms, thrive under a specific range of environmental conditions
or optima and these optima are different for each species. For this reason, diatom
assemblages are excellent paleoceanographic indicators (Smol, 2002). Table 1 delineates
which species were grouped together into specific environmental niches. Our
interpretations of the paleoceanographic sea surface conditions at the Bering Sea shelf-
slope break during MIS 12 to 10 are based on changes in these 8 groups and the
variability of *Neodenticula seminae*, an indicator of the Alaskan Stream and North
Pacific water (Katsuki and Takahashi, 2005; Sancetta, 1982) (Fig. 7).

### 376  4.2.2.1    Sea Ice Species

Epontic diatoms are those that bloom attached to the underside of sea ice or within brine
channels in the ice. This initial bloom occurs below the ice as soon as enough light
penetrates to initiate photosynthesis in the Bering Sea, which can occur as early as March
(Alexander and Chapman, 1981). The centric diatom, *Melosira arctica,* and pennate



diatoms, *Nitzschia frigida* and *Navicula transitrans* are among the major components of
the epontic diatom bloom (von Quillfeldt et al., 2003) and all are found in the sediments
at U1345A, although they tend to be quite rare.
A second ice-associated bloom occurs as sea ice begins to break up on the Bering Sea
shelf. This bloom is referred to as the marginal ice zone bloom and many of its members
are common species in the sediment assemblage including the pennate diatom,
*Staurosirella* cf. *pinnata* (=*Fragilaria* cf. *pinnata*)*,* and the centric diatoms, *Bacterosira*
*bathyomphala* and several *Thalassiosira* species including *Thalassiosira antarctica*
(Schandelmeier and Alexander, 1981; Shiga and Koizumi, 2000; von Quillfeldt et al.,
2003)*. T. antarctica* resting spores have been classified in various ways in the past and
their ecology is not well understood. However, *T. antarctica* is a member of the marginal
ice zone flora (von Quillfeldt et al., 2003) and was the only organism found in thick pack
ice (Horner, 1985). The resting spores are associated with coastal or ice-margin waters
that range from −1 to 4° C and have relatively low salinity (25–34‰) (Barron et al.,
2009; Shiga and Koizumi, 2000). In Antarctica, *T. antarctica* blooms in concert with
frazil and platelet ice growth in the fall (Pike et al., 2009). This same association has not
yet been observed in the Arctic, though it is a possibility. High abundances might indicate
that ice formed early enough in the fall that light and/or nutrients were high enough to
support *T. antarctica* growth then.
Several diatom species are present in both types of sea ice blooms, and so while they are
indicators of ice presence, they cannot be used to distinguish between types of sea ice.
These species are grouped under "both ice types" and include such common diatoms as
*Fragilariopsis oceanica, Fragilariopsis cylindrus, Fossula arctica,* and many Naviculoid
pennate diatoms (Saito and Taniguchi, 1978; Sancetta, 1981; Schandelmeier and
Alexander, 1981; von Quillfeldt, 2001; von Quillfeldt et al., 2003).
Epontic species are present in low relative percent abundances (< 5%) throughout much
of the record, but there is a marked absence of them during the laminated interval from
423 to 410 ka (129.96-126.45 mbsf). Marginal ice zone species fluctuate between 4% and
14% throughout the record and do not show any trends in abundance changes. The





grouping of species found both within the ice and in the water surrounding ice, however,
is also somewhat reduced during laminated intervals (Fig. 7).

### 4.2.2.2  Warmer Water Species

Diatoms associated with warmer water or classified as members of temperate to
subtropical assemblages are rare in this record; however, they are present. This group
includes *Azpeitia tabularis, Thalassiosira eccentrica, Shionodiscus oestrupii*
(=*Thalassiosira oestrupii*)*,* and *Thalassiosira symmetrica.* (Fryxell and Hasle, 1972;
Lopes et al.; Sancetta; Sancetta and Silvestri, 1986).
Relative abundances of warmer water species are quite low throughout the record (<5%),
and are highest (3-4%) during mid to late MIS 11 approximately ~410 to 391 ka (126.74
to 116.50 mbsf) (Fig. 7).

### 4.2.2.3  Alaskan Stream Species

*Neodenticula seminae* is often used as a tracer of North Pacific water, in particular the
Alaskan Stream (e.g. Caissie et al., 2010; Katsuki and Takahashi, 2005). But its
distribution also varies on glacial-interglacial time scales within the Pacific Ocean
(Sancetta and Silvestri, 1984). It is adapted to the low productivity of the North Pacific
gyre and is heavily silicified which could lead to high proportions of *N. seminae*
reflecting simply dissolution of finely silicified diatoms (Sancetta, 1982, 1981). *N.*
*seminae* is used here as a tracer of Pacific water with the above caveats.
Absolute abundances of *N. seminae* began to increase at 422 ka as global eustatic sea
level rises above -50 mapsl. Abundance then decreases slowly over the course of the
Termination V Laminations and peaks again at 392 ka and 382 ka. As sea level drops
below -50 mapsl, *N. seminae* is no longer present at U1345. Relative percent abundances
remain stable at ~2% relative percent abundance between 422 and 400 ka (129.62-123.62
mbsf), then peaks at 13% at 392 ka (121.22 mbsf) (Fig. 6). Low proportions of *N.*
*seminae* during the Termination V Laminations are likely due to the overwhelming
proportion of *Chaetoceros* RS during this time.



### 4.2.2.4    High Productivity Species

*Chaetoceros* resting spores are the dominant taxa included in the high productivity group.
*Chaetoceros* RS have been used as indicators of high productivity (e.g. Caissie et al.,
2010) and are often found in locations influenced by intense upwelling (Lopes et al.,
2006; Sancetta, 1982). In addition, *Chaetoceros socialis* can be a common member of the
marginal ice zone bloom (von Quillfeldt, 2001) and a dominant member of the sub ice
bloom (Melnikov et al., 2002). *Chaetoceros furcellatus* is also associated with the
marginal ice zone bloom (von Quillfeldt, 2001). Unfortunately, the morphology of
*Chaetoceros* resting spores is quite variable, and they cannot be classified definitively
without the more labile vegetative cell also present (Tomas, 1996). *Odontella aurita,*
*Thalassionema nitzschioides* are also included in the high productivity group although
they are also associated with the marginal ice zone (von Quillfeldt et al., 2003), areas of
high productivity (Aizawa et al., 2005), and upwelling (Lopes et al., 2006; Sancetta,
1982). It should be noted that we can not discern between high productivity due to
upwelling and high productivity due to other factors because the diatom proxies are not
sufficiently refined to distinguish between the two.
It may be that the combination of upwelling and ice melt at the shelf slope break in the
Bering Sea is responsible for correlation between these two environmental niches. The
spring-blooming *Thalassiosira pacifica* and small (<10 μm) *Thalassiosira* species round
out the high productivity group due to their associations with high productivity and
upwelling specifically in the Bering Sea and North Pacific (Aizawa et al., 2005; Hay et
al., 2007; Katsuki and Takahashi, 2005; Lopes et al., 2006; McQuoid and Hobson, 2001;
Saito and Taniguchi, 1978).
Like *Chaetoceros* RS, high productivity species mimic the trend of the insolation curve
(Berger and Loutre, 1991) with highest relative abundances (60-70%) occurring during
high levels of insolation (Fig. 7). The lowest relative abundances (15-20%) of high
productivity species occur between 403 and 390 ka (124.21 to 120.07 mbsf) when both
obliquity and insolation are low. High productivity species are high during both the
Termination V Laminations and during the late MIS 11 laminations (Fig. 7).




**4.2.2.5        Dicothermal Water Indicators and Late Summer Species**

A cold layer of water found between seasonally warmer surface and warmer deep water characterizes dicothermal water. It is stable because of its very low salinity. In the Sea of Okhotsk and the Bering Sea, the dicothermal layer is often associated with melting sea ice. The highest abundances of *Shionodiscus trifultus* are found associated with this highly stratified, cold water in the Sea of Okhotsk today (Sancetta, 1981; Sancetta and Silvestri, 1986).

*Actinocyclus curvatulus* has been observed living in water surrounding sea ice (von Quillfeldt et al., 2003); however, it is neither a common member of the marginal ice zone flora, nor is its spatial distribution in the Bering Sea consistent with the distribution of sea ice (Sancetta, 1982). Its relative percent abundances are more closely associated with those of *S. trifultus* (Sancetta, 1982)*,* and so it was grouped with *S. trifultus* as an indicator of dicothermal water.

Genera present in the Bering Sea during late summer (*Coscinodiscus, Leptocylindrus,* and *Rhizosolenia*) (Aizawa et al., 2005; Lopes et al., 2006; von Quillfeldt et al., 2003) tend to co-vary with the dicothermal water indicators, so the two groups were merged for comparison with other diatom groups.

These two groups are highest (18% relative abundance) at ~401 ka (123.62 mbsf) as insolation declines. This peak is coeval with the peak in fresh water species and an intermediate peak in *N. seminae* and occurs immediately following a peak in neritic species. Dicothermal water indicators and summer species are lowest (< 1%) during the Termination V Laminations (~424-412 ka). Intermediate relative abundances (1% to 5%) occur during MIS 12 and above 392 ka (121.04 mbsf) (Fig. 7).

493

**4.2.2.6        Shelf to Basin Transport Indicators**

Freshwater species are rare, but present in the record. They include the centric species *Lindavia* cf. *ocellata* and *L. radiosa* (Håkansson, 2002). Additional freshwater diatoms found in the record are species also found in sea ice (*S.* cf. *pinnata*) (von Quillfeldt et al.,



2003) or in the neritic zone (*Cyclotella stylorum)* (Barron et al., 2009) and so these species were placed in the marginal ice zone and neritic groups respectively.

The dominant species in the neritic group is *Paralia sulcata,* which is an interesting species because it can be either planktic or benthic (Kariya et al., 2010) and is associated both with river deltas and the species *Melosira sol* (Sancetta, 1982). It can be a member of the marginal ice zone assemblage (von Quillfeldt et al., 2003) though Pushkar (1999) asserts that *P. sulcata* indicates water shallower than 20 m. Its high abundances in Bering Strait may mean that it is adapted to moving water (Sancetta, 1982). *P. sulcata* thrives in water that is warmer than 3 degrees (Zong, 1997), with low light (Blasco et al., 1980) and low salinity (Ryu et al., 2008).

The fresh water group is notably absent from much of the core, but prevalent between 401 and 392 ka (123.70 mbsf and 121.20 mbsf); it reaches its highest relative percent abundance (12%) at 401 ka (123.62 mbsf). Neritic species, on the other hand maintain ~10% relative abundance throughout the core. They are lowest during the Termination V Laminations and increase dramatically around 404 ka (124.61 mbsf) to almost 50% of the assemblage (Fig. 7).

## 4.3 Calcareous Nannofossils

Calcareous nannofossils were examined between 432-405 ka (133.4 to 125.0 mbsf); one third of the samples were barren and only one sample (418 ka; 128.8 mbsf) had sufficient individuals to estimate relative percent abundances (Fig. 7). This sample is located midway through the Termination V Laminations when the diatom assemblage is overwhelmingly dominated by *Chaetoceros* RS. Small *Gephyrocapsa* dominates (>50%) the calcareous nannofossil assemblage. There are 35% medium *Gephyrocapsa,* 9% *Coccolithus pelagicus*, and 1% *G. oceanica*.



## 4.4 Geochemistry

### 4.4.1 Organic and Inorganic Carbon Content

Total organic carbon (TOC) roughly follows the trend of relative percent abundances of *Chaetoceros* RS, with higher values during the Termination V Laminations. Mean TOC value during MIS 12 is 0.76%, and during the Termination V Laminations, it is 1.11%. TOC decreases temporarily in sync with depleted $\delta^{15}N$ values, before rising linearly from 404 ka (124.77 mbsf) to 374 ka (115.39 mbsf). TOC is again high during the late MIS 11/MIS 10 laminations.

In contrast, inorganic carbon, calculated as % $CaCO_2$ is less than 1% for most of the record; however, it increases up to 3.5% during the laminated intervals and also at 382 ka (117.87 mbsf), 392 ka (110.00 mbsf), and 408 ka (125.82 mbsf).

### 4.4.2 Terrigenous Input Indicator (C/N)

The ratio, C/N is one of two proxies used as indicators of marine versus terrigenous organic matter, with marine values typically ranging from 5-7 and terrigenous ratios over 20 (Meyers, 1994; Redfield et al., 1963).

Throughout the record, C/N indicates primarily a marine source for organic matter. During MIS 12, C/N is highly variable, when sea level is below -50 m apsl. As sea level rises during Termination V, C/N values increase from 6 to more than 9. The highest C/N value occurs at the start of the Termination V Laminations. C/N decreases as sea level rises until at 400 ka (123.62 mbsf) it stabilizes near 7 for the remainder of the record.

### 4.4.3 Bulk Sedimentary Stable Isotopes

#### 4.4.3.1 Carbon Isotopes

Stable isotopes of carbon are also used as an indicator of marine vs. terrigenous organic matter with $\delta^{13}C$ values near -27 indicating C3 plant-sourced organic matter; values between -22 and -19 are typical for Arctic Ocean marine phytoplankton and -18.3 is average for ice-related plankton (Schubert and Calvert, 2001). However, it has been





shown that $\delta^{13}$C is sometimes related more to growth rate, cell size, and cell membrane
permeability, so it may reflect changing phytoplankton groups instead of simply marine
vs. C3 plant sources of organic matter in U1345.
Carbon isotopic values range between -22 ‰ and -26 ‰ and are generally anticorrelated
with C/N values. These values indicate a mix of marine phytoplankton and C3 plants as
the main contributors to organic matter at the site. At the onset of the Termination V
Laminations, $\delta^{13}$C becomes more negative and then gradually increases to a maximum of
-22.33 at 404 ka (124.62). After 400 ka (123.5 mbsf), $\delta^{13}$C is relatively stable around -
23.5‰.

**4.4.3.2     Nitrogen Isotopes**
Nitrogen, in the form of nitrate, is a key nutrient for phytoplankton growth. Diatoms
preferentially assimilate the lighter isotope, $^{14}$N, which in turn enriches surface waters
with respect to $^{15}$N (Barron et al., 2009; Shiga and Koizumi, 2000). Keeping in mind the
effects of nitrification of oxygen rich and poor sediments (Brunelle et al., 2007), the
efficiency of nitrogen utilization can be estimated by examining the $^{15}$N/$^{14}$N ratio of
nitrogen in either bulk sedimentary organic matter, with enriched values of $\delta^{15}$N
indicating higher nutrient utilization. Sponge spicules (very low $\delta^{15}$N values) and
radiolarians (highly variable $\delta^{15}$N values) may contaminate the $\delta^{15}$N of bulk organic
matter, however we looked for and found no correlation between spicule abundance and
$\delta^{15}$N in our samples.
Surprisingly, $\delta^{15}$N is relatively stable throughout the study interval, fluctuating around an
average value of 6.4‰, though there are several notable excursions. Coeval with sea level
rise and increased relative percent *Chaetoceros* RS, $\delta^{15}$N decreased 2.7‰ to 4.4‰ before
recovering to average values during the Termination V Laminations. Two other
depletions occur at 405 ka (124.77 mbsf) and 393 ka (121.62 mbsf), the first is the most
extreme and reaches 2.9‰.



## 5    Discussion

The study interval can be broken into five zones based on changes in diatom assemblages and lithology (Fig. 7): MIS 12, Termination V, Peak MIS 11, Beringian Glacial Initiation, and Late MIS 11. These zones reflect changing sea ice, glacial ice, sea level, and SST and correspond to events recognized elsewhere in ice cores and marine and lake sediments.

## 5.1 Marine Isotope Stage 12 and Early Deglaciation (beginning of record to 425 ka)

The beginning of the record to 425 ka chronicles conditions at the end of MIS 12. Although diatom accumulation rate is quite low, a relatively diverse assemblage characterizes this period (Fig. 5) with moderate amounts of sea ice, high productivity, and dicothermal species (Fig. 7), indicating seasonal sea ice with highly stratified waters during the ice-melt season. Nitrogen isotopes indicate high nutrient utilization (Fig. 7) consistent with nitrogen-limited productivity in stratified waters. Similarly, the North Atlantic was also highly stratified with significantly reduced NADW production (Poli et al., 2010). High stratification appears to have led to lowered productivity in both the Atlantic and Pacific.

Sea level was low during this interval (Rohling et al., 2010), placing U1345 proximal to the Beringian coast (Fig. 2). With the Beringian shelf exposed, the continent was relatively cold and arid (Glushkova, 2001). In western Beringia, Lake El'gygytgyn was perennially covered with ice, summer air temperatures were warming from 4 to 12° C and annual precipitation was low (200-400 mm) (Vogel et al., 2013).

Numerous shell fragments, two sand layers and the highest percentages of clay-sized sediments in the record were deposited during MIS 12 (Figs 4 and 8) indicating input of terrigenous material, however, both C/N and $\delta^{13}$C indicate that the organic matter during MIS 12 is of marine origin. Non-organic, terrigenous material could be transported to U1345 by large glacial meltwater rivers, icebergs, or sea ice. It is unlikely that meltwater rivers played a large role in sediment transport at this time because terrigenous organic matter and fresh water diatoms are absent and there are only moderate amounts of



diatoms transported from shallow waters (Fig. 8). This may also reflect the reduced area
of submerged continental shelf. In addition, glacial ice was likely restricted to mountain-
valley glaciers, similar to the last glacial maximum (e.g. Glushkova, 2001). These small,
distant glaciers would not have produced large amounts of ice bergs though occasional
glacial ice rafted debris (IRD) may have come from the Koryak Mountains, Aleutians or
Beringia. Consistent with this, sediments typical of glacial IRD, such as dropstones, are
sparse. Terrigenous materials found in MIS12 sediments are most likely evidence of sea
ice rafting which tends to preferentially entrain clay and silt (Reimnitz et al., 1998) or of
minor downslope transport. In contrast, the North Atlantic is surrounded by ice sheets
readily calving ice bergs and MIS 12 is characterized as an intense glacial period with ice
rafted debris found as far south as Bermuda (Poli et al., 2000). Evidence for warming and
the reduction of IRD begins as early as 430 ka in the North Atlantic (Kandiano et al.,
2012) and the strength of the Gulf Stream increases in step with this glacial ice loss
(Chaisson et al., 2002).

**624 5.2 Early MIS 11: Termination V (425-423 ka)**

Termination V is the transition from MIS 12 to MIS 11. Worldwide, it is a rapid
deglaciation that is followed by a long (up to 30 kyrs) climate optimum (Milker et al.,
2013). At U1345, it can be broken into two stages, the first part from 425-423 ka, and the
second part from 423-410 ka, which is notably dominated by laminated sediments and is
discussed in the next section. The first part of Termination V corresponds with a local
maxima in insolation at 65°N (Schimmelmann et al., 1990) and increasing temperatures
in Antarctica (EPICA community members, 2004), the North Atlantic (Voelker et al.,
2010), and globally (Milker et al., 2013).
At U1345, the first part of Termination V is expressed as increasing absolute diatom
abundances and relative percent abundance of *Chaetoceros* RS while sea ice diatoms and
$\delta^{15}N$ values decrease (Fig. 7) indicating gradually increasing productivity coupled with
decreasing nutrient utilization and sea ice.
At the same time, episodic increases in productivity were occurring in places as distant as
Lake Baikal (Prokopenko et al., 2010) and the North Atlantic (Chaisson et al., 2002;



Dickson et al., 2009; Poli et al., 2010) and NADW formation was intensifying (Poli et al.,
2010). Ventilation of NADW generally continues to increase from 424 to 410 ka to its
strongest and then weakens over the course of the interglacial (Thunell et al., 2002). In
addition, the flux of terrigenous dust was decreasing near Antarctica reflecting perhaps a
decrease in the strength of Southern Ocean winds (Wolff et al., 2006). Evidence for
higher productivity in the Bering Sea, possibly caused by intensified upwelling, suggests
teleconnections between NADW formation, the strength of the southern winds, upwelling
in the North Pacific, and northward flow through Bering Strait (e.g. DeBoer and Nof,

647    2004).


### 5.2.1   Early MIS 11, part 2: Laminations (423-410 ka)

The most prominent expression of Termination V is a 3.5 m thick laminated interval
deposited beginning at 423 ka (Fig. 7) when insolation was high at 65°N (Berger and
Loutre, 1991). Its presence indicates that the bottom water at 1,000 m in the Bering Sea
was dysoxic for more than 11 kyrs. These laminations are characterized as Type I
laminations with a notably high diatom content (Fig. 4). Several lines of evidence point
towards high productivity among multiple phytoplankton groups as opposed to simply a
change in preservation. There is a two orders of magnitude increase in diatom
abundances since MIS 12, a low-diversity diatom assemblage dominated by *Chaetoceros*
RS, an abrupt increase in percent organic carbon, high percent $CaCO_3$ and abundant
calcareous nannofossils dominated by small *Gephyrocapsa*. Furthermore, enriched $\delta^{15}N$
values indicate increased nitrogen utilization that likely fed this increased productivity
(Fig. 7).
Sea ice is reduced during this interval with almost no epontic diatoms present and
reduced amounts of other sea ice diatoms (Fig. 7). The $\delta^{13}C$ pattern of depleted values at
the start of the laminated interval and increasingly enriched values until about 409 ka is
very similar to the $C_{org}/N$ story (Fig. 7) reflecting the highest contribution from C3 plant
organic matter at the onset of the laminated interval as the tundra-covered Bering Shelf is
flooded. However, the depletion in $\delta^{13}C$ during the Termination V Laminations occurs at



the same time that *Chaetoceros* RS overtake the assemblage (Fig. 7), so a species effect
cannot be ruled out.
The diatom record, on the other hand has the lowest contribution of neritic diatoms
during the laminated interval and virtually no fresh water diatoms (Fig. 7), suggesting
that although terrigenous organic matter was an important input at the site, coastal, river,
or swamp/tundra diatoms were not major constituents of this terrigenous organic matter.
The sum of this evidence of high productivity, reduced sea ice, and terrigenous input is
similar to changes in productivity in this region during Termination I (Brunelle et al.,
2007; Caissie et al., 2010). At the start of Termination I, productivity initially increased
while nitrogen utilization decreased, then an abrupt increase in productivity and nitrogen
utilization was recorded (Brunelle et al., 2007; Brunelle et al., 2010). It is plausible that
increased nitrogen availability drove higher primary productivity as floods scoured fresh
organic matter from the submerging continental shelf (Bertrand et al., 2000). Rapid input
of bioavailable nitrogen as the shelf was inundated has been suggested to explain
increasing productivity during the last deglaciation in the Sea of Okhotsk (Shiga and
Koizumi, 2000) and during MIS 11 in the North Atlantic (Poli et al., 2010) and also may
have contributed to dysoxia by ramping up nutrient recycling, bacterial respiration, and
decomposition of organic matter in the Bering Sea.
The brief nitrogen utilization decrease just prior to the laminations (Fig. 7), suggests that
productivity was limited by some other factor, such as light or micronutrients, and could
not increase proportional to the increase in available nitrogen. Lam (2013) suggests that
during the last deglaciation, a breakdown in stratification limited productivity by creating
a very deep mixed layer that extended below the photic zone. This seems possible during
Termination V, since diatom indicators for stratified waters (dicothermal species) and
epontic diatoms decline coeval with the increase in productivity indicators (Fig. 7),
though seasonal sea ice remains and likely provides a mechanism for maintaining
stratification to some extent. As the interglacial began however, we would expect this
light limitation to be removed when stratification was re-established. However, if
dicothermal diatoms are indicators for stratification, then stratification is not re-
established until long after $\delta^{15}$N values increase (Fig. 7), suggesting that if there is a limit



on productivity during the early deglacial it is likely not light via a deep mixed layer (Obata et al., 1996). In contrast, a nearby core (HLY 0202 JPC3) displayed laminated sediments for only about 500 years during the last deglaciation (Cook et al., 2005), suggesting that the two Terminations were very different.

There is a "Younger Dryas-like" temperature reversal seen midway through Termination V in the North Atlantic (Voelker et al., 2010), Antarctica (EPICA community members, 2004) and at Lake El'gygytgyn (Vogel et al., 2013), however there is no evidence for such an event in the Bering Sea.

## 5.3 Peak MIS 11 (423-394 ka)

Globally, peak interglacial conditions (often referred to as MIS 11.3 or 11c) are centered around 410 ka, though the exact interval of the temperature optimum varies and lasted anywhere from 10 to 30 kyrs (Kandiano et al., 2012; Kariya et al., 2010; Milker et al., 2013). At U1345, peak interglacial conditions begin during the Termination V Laminations and continue until 394 ka.

Both decreasing $C_{org}/N$ and increasing $\delta^{13}C$ indicate that input of terrigenous organic matter decreases from the onset of the Termination V Laminations until mid MIS 11 (400 ka) at which time the organic matter remains solidly marine sourced for the remainder of the record (Fig. 7). Sea level is high and the Pacific water indicator, *N. seminae*, is found at the site beginning at 424 ka.

Throughout MIS 11, *Chaetoceros* RS, a species indicative of high productivity, is generally higher when insolation is higher and lower when isolation is lower (390-404 ka; Fig. 7). However, although their fluctuations are small, warm water species show the opposite trend, with higher proportions of warm water diatoms when insolation is low (Fig. 7). If higher proportions of warm water diatoms indicate warmer water, then this suggests that productivity is highest in colder waters but when insolation is high, and lowest in warmer waters when insolation is low.

During MIS 11c, global ice volume was the lowest that it has been for the past 500 kyrs (Lisiecki and Raymo, 2005), and generally continental temperatures were warmer than



today (D'Anjou et al., 2013; de Vernal and Hillaire-Marcel, 2008; Lozhkin and Anderson, 2013; Lyle et al., 2001; Melles et al., 2012; Pol, 2011; Prokopenko et al., 2010; Raynaud et al., 2005; Tarasov et al., 2011; Tzedakis, 2010; Vogel et al., 2013) with a northward expansion of boreal forests in Beringia (Kleinen et al., 2014). However, it was not warm uniformly world-wide. At U1345, the relative percent warm water species suggest that SSTs during MIS 11c were only slightly warmer than during MIS 12. Indeed, MIS 11 is not the warmest interglacial in most marine records (Candy et al., 2014). This is especially evident in the Nordic Seas where MIS 11 SSTs were lower than Holocene values, although no IRD was deposited between 408 and 398 ka (Bauch et al., 2000).

However, MIS 11c was very humid in many places. In the Bering Sea, modeling studies estimate up to 50 mm more precipitation than today at 410 ka (Kleinen et al., 2014). The most humid, least continental period recorded in the sediments at Lake Baikal occurs from 420-405 ka (Prokopenko et al., 2010), and extremely high precipitation are recorded at Lake El'gygytgyn on the nearby Chukotka Peninsula from 420-400 ka (Melles et al., 2012). Conditions in Africa during MIS 11c were similar to the Holocene African humid period. In addition, pollen records from Western Europe also reflect humid environments (Candy et al., 2014). A warmer, moister climate in Western Europe and Africa is indicative of increased Atlantic Meridional Overturning Circulation (AMOC) (Bauch, 2013). AMOC appears to be stable over MIS 11 (Milker et al., 2013) as evidenced by high carbonate in the North Atlantic (Chaisson et al., 2002; Poli et al., 2010). Interestingly, small carbonate peaks in the Bering Sea are contemporaneous with those on the Bermuda Rise, suggesting teleconnections between the two regions (Fig. 7). These conditions are similar to a modern day negative North Atlantic Oscillation (NAO) which is linked to wet conditions in N. Africa, weaker westerlies, more zonal storm tracks, a dry Northern Europe, colder Nordic Seas and increased sea ice in the North Atlantic (Kandiano et al., 2012).

### 5.3.1 Bering Strait Current Reversal (406-402 ka)

Between 405 and 394 ka, there is an unusual diatom assemblage and grain size distribution at Site U1345. There are several possible explanations for deposition of



shallow water and fresh water species along with large changes in sediment grain size.
We will consider two possibilities in detail: Bering Strait current reversal and glacial
surge in Beringia.
On St. Lawrence Island in the Northern Bering Sea (Fig. 1), evidence for Arctic mollusks
entering the Gulf of Anadyr suggests that flow through Bering Strait was reversed at
some point during the Middle Pleistocene (Hopkins, 1972). Unfortunately, this event is
poorly dated. If Bering Strait flow were reversed due to a meltwater event (DeBoer and
Nof, 2004), we would expect a temporary reduction in NADW formation and an increase
in southerly winds from Antarctica (DeBoer and Nof, 2004). In the Bering Sea, we would
expect to see an increase in common Arctic or Bering Strait diatom species and a
decrease in North Pacific indicators. In addition, the clay minerals in the Arctic Ocean are
overwhelmingly dominated by illite (Ortiz et al., 2012), which is a clay mineral that tends
to adsorb large amounts of ammonium (Schubert and Calvert, 2001). So, if net flow were
to the south, one might also expect to find decreased $C_{org}/N$ and $\delta^{15}N$ values resulting
from increased illite deposition.
A warm Arctic Ocean during MIS 11 suggests increased Pacific water input through
Bering Strait (Cronin et al., 2013). Proxy evidence for NADW ventilation (decreases in
$CaCO_3$ % and $\delta^{13}C$) indicates that between 412 and 392 ka, NADW formation decreased
for short periods (< 1 ka) (Poli et al., 2010). In contrast, AABW formation appears to
have drastically slowed around 404 ka, suggesting that winds derived from Antarctica
decreased as opposed to increased (Hall et al., 2001).
At U1345, $C_{org}/N$ values began decreasing linearly starting at 409 ka, productivity
sharply decreases at 406 ka, $\delta^{15}N$ values are the most depleted at 405 ka, just 1 kyr before
a conspicuous peak in *P. sulcata,* a common diatom in the Bering Strait. Finally, diversity
is highest around 400 ka, due to the multiple contributions of Arctic species (fresh water,
shelf, coastal, sea ice) and common pelagic diatoms, while the North Pacific indicator, *N.*
*seminae* maintains low relative abundances throughout this interval. The sum of this
evidence does point toward species migration from the Arctic Ocean southward.
However, these changes occur in series over 4 kyrs or more and there is no synchronicity
between NADW formation and Antarctic winds. Therefore, there is no consensus in the



evidence to support or reject the hypothesis of reversed flow through Bering Strait during
MIS 11.

### 5.3.2  Glacial Inception in Beringia (405-394 ka)

Regardless of whether flow through Bering Strait reversed during peak MIS 11, the
interval between 405 and 394 ka contains an unusual diatom assemblage and grain size
distribution. Diatom assemblages are similar to that found in sediments from the
Anvillian Transgression 800 km northeast of U1345 near Kotzebue (Fig. 1) (Pushkar et
al., 1999). In the Bering Sea, a large peak in neritic species occurs at 404 ka followed by
the highest relative percentages of fresh water species at the site and a slight increase in
sea ice diatoms from 400 to 394 ka (Fig. 7).
Despite the deposition of shallow and fresh water species, the proportion of marine to
terrestrial carbon was the highest in the entire interval. However, primary productivity
was quite low during this interval with high nitrogen utilization reflected in the $\delta^{15}$N
values. Two large depletions in $\delta^{15}$N bracket this interval and occur as *Chaetoceros* RS
decrease in relative percent abundance, but only the older depletion is also associated
with a decrease in the number of diatom valves per gram of sediment (Fig. 8). The older
depletion may reflect an environment that is limited by micronutrients such as iron as sea
level approaches its maximum.
Detailed grain size analysis shows a trend of increasing clay sized grains as well as a
broad increase in sand sized grains and in particular grains greater than 250 μm (Fig. 8).
All samples are poorly to very poorly sorted (See Supplemental Material). In addition,
shipboard data shows an increase in the presence of large, isolated clasts > 1 cm in
diameter, a cluster of sand layers (Fig. 8), and a thick interval of silty sand (Takahashi et
al., 2011) around 411 ka (Fig. 4).
The sum of this evidence leads us to propose that the interval highlighted in grey on
Figure 8, reflects a glacial advance that may be the onset of the Nome River Glaciation at
~404 ka. This advance is short-lived in the Bering Sea and is followed by a period when
intensified winds blew fresh water diatoms more than 1000 km off shore to Site U1345.



Glacial ice is effective at carrying terrigenous and near shore particles far from land.
Previous work has suggested that sediments deposited by icebergs should be poorly
sorted and skew towards coarser sediments (Nürnberg et al., 1994). Sediments greater
than 150 μm are likely glacially ice rafted (St. John, 2008), however it is not possible to
distinguish sediments deposited by glacial versus sea ice on grain size alone (St. John,
2008). Both types of ice commonly carry sand-sized or larger sediments (Nürnberg et al.,
1994). Sea ice diatoms should not be found in glacial ice, instead, we would expect
glacial ice to be either barren, or to carry fresh water diatoms from ice-scoured lake and
pond sediments.
The Nome River Glaciation is the most extensive glaciation in central Beringia and is
dated to Middle Pleistocene. Although it has not been precisely dated, it is likely
correlative with MIS 11 (Kaufman et al., 1991; Miller et al., 2009). Nome River
glaciomarine sediments are found in places such as St. Lawrence Island (Gualtieri and
Brigham-Grette, 2001; Hopkins, 1972), the Pribilof Islands (Hopkins, 1966), the Alaska
Arctic coastal plain (Kaufman and Brigham-Grette, 1993), Kotzebue (Huston et al.,
1990), Nome (Kaufman, 1992), and Bristol Bay (Kaufman et al., 2001) (Fig. 1). Mollusks
and pollen in these sediments reflect a tundra environment with temperatures similar to
today (Hopkins, 1972; Kaufman and Brigham-Grette, 1993) or warmer than today
(Pushkar et al., 1999) with significantly reduced or absent sea ice (Kaufman and
Brigham-Grette, 1993; Pushkar et al., 1999). These sites all contain evidence that glaciers
in Beringia advanced, in some cases more than 200 km, and reached tidewater while
eustatic sea level was high (Huston et al., 1990).
Although global sea level was near its maximum, and much of the world was
experiencing peak MIS 11 conditions (Candy et al., 2014), there is evidence that the high
latitudes were already cooling. At 410 ka, insolation at 65° N began to decline (Berger
and Loutre, 1991), cooling began at 407 ka in Antarctica, expressed both isotopically and
as an expansion of sea ice (Pol, 2011). Millennial scale cooling events are recorded at
Lake Baikal (Prokopenko et al., 2010). By 405 ka, there is some evidence globally for ice
sheet growth (Milker et al., 2013) as Lake Baikal begins to shift towards a dryer, more
continental climate (Prokopenko et al., 2010) and productivity declines at Lake
El'gygytgyn (Melles et al., 2012). Around 400 ka, SSTs decrease temporarily by 2° C in



the Arctic Ocean (Medeleev Ridge) (Cronin et al., 2013) and permanently at Lake
El'gygytgyn (D'Anjou et al., 2013). Precipitation also decreases at Lake El'gygytgyn
(Melles et al., 2012). Modelling results show that by 400 ka, the Bering Sea is expected
to have temperatures cooler than today with increased sea ice (Kleinen et al., 2014).
We suggest that Beringian glaciation during MIS 11 was initiated ~404 ka by decreasing
insolation when eccentricity was high and perihelion coincided with the equinox
(Schimmelmann et al., 1990). Solar forcing coupled with a proximal moisture source, the
flooded Beringian shelf, to drive snow buildup (Huston et al., 1990; Pushkar et al., 1999)
and glacial advance. Precipitation at Lake El'gygytgyn, just west of the Bering Strait, was
two to three times higher than today (Melles et al., 2012). A similar "snow gun"
hypothesis has been invoked for other high latitude glaciations (Miller and De Vernal,
1992); however, Beringia is uniquely situated. Once sea level began to drop, Beringia
became more continental and arid (Prokopenko et al., 2010) and the moisture source for
these glaciers was quickly cut off.
In central Beringia, glaciers from coastal mountains on chukotka advanced to St.
Lawrence Island and glaciers from the western Brooks Range advanced into Kotzebue
Sound as global eustatic sea level dropped coincident with decreased insolation during
Northern Hemisphere summers (Berger and Loutre, 1991), Lake El'gygytgyn returned to
glacial conditions by 398 ka, and globally MIS 11.3 ended (Milker et al., 2013; Poli et al.,
2010; Voelker et al., 2010).

**5.3.3  Alternative Explanations**
There are several other explanations for how these sediments could have been carried
more than 300 km from the coast out over the shelf-slope break and deposited in 1000 m
of water: turbidites or strong density currents on the shelf, sediment reworking and
winnowing, sea ice transport, and eolian deposition
The location of Site U1345 on a high interfluve minimizes the likelihood that sediments
will have been transported and deposited here by turbidites or other down-slope currents.
Sancetta and Robinson (1983) argue that benthic pennate species were transported out of





shallow water by rivers and turbidity currents during glacial periods. They do not consider ice as a transport mechanism (Sancetta and Robinson, 1983). If turbidites were present, we would expect fining up sequences in the detailed grain size analysis, slumping or distorted sedimentation in the core and clear erosive surfaces. But there is no evidence of turbidite deposition (Takahashi et al., 2011). If winnowing were a dominant transport mechanism, the sediments should be well sorted. Instead, the presence of multiple terrigenous grain sizes indicates that the sediments are relatively poorly sorted and the Folk and Ward method (Blott and Pye, 2001) classifies all samples as either poorly sorted or very poorly sorted (See supplemental material).

Sea ice could bring neritic (though probably not freshwater) diatoms out to deeper waters as it preferentially entrains silt and clay size particles (Reimnitz et al., 1998). However, if there was an increase in sea ice, we would expect to see a significant increase in sea ice diatoms during this interval. Instead we see only a small increase in sea ice related species, primarily epontic species. Additionally, during this time, the marginal ice zone assemblage is dominated by *T. antarctica* RS which is a taxon primarily found in coastal, low salinity areas (Barron et al., 2009; Shiga and Koizumi, 2000), so its presence may be further support for increased shelf to basin transport.

Eolian deposition of diatoms is a common event in Antarctica where strong katabatic winds transport mainly small (up to 50 μm in diameter), non-marine diatoms (McKay et al., 2008). The freshwater diatoms that are abundant between 409 and 405 ka are dominated by species that tend to be quite small. *Lindavia* cf. *ocellata* ranges from 8-20 μm and *Lindavia radiosa* from 7-35 μm. Wind-driven deposition of these species is the most probably explanation for their transport more than 800 km from shore, therefore, this interval may represent a period of time when northerly winds intensified over Beringia.

## 5.4 Late MIS 11 (younger than 394 ka)

After 394 ka, upwelling indicators are the lowest in the record and linearly increase to the top of the record. This is in contrast to a slight increase in diatom abundance, which increases at 393 ka and then remains relatively stable to the top of the record. Sea ice



indicators also remain relatively high from 392 to the top of the record and dicothermal
species reflect moderately stratified waters. Warm water species decrease from 390 ka to
the top of the record (Fig. 7). The sum of this evidence indicates that at the end of MIS
11, summers were warm and sea ice occurred seasonally, perhaps lasting a bit longer than
at other times in the record. Modelling results indicate that at 394 ka, temperatures were
below modern by 0° to 2° C, and precipitation in Beringia was relatively low, like today
(Kleinen et al., 2014). These patterns reflect general cooling worldwide (de Abreu et al.,
2005; Prokopenko et al., 2010; Raynaud et al., 2005). IRD is again deposited in the North
Atlantic beginning around 390 ka.
Eustatic sea level decreased beginning about 402 ka (Rohling et al., 2010), but sea level
was high enough though to allow *N. seminae* to reach the shelf slope break until about
380 ka (Fig. 7). As sea level dropped, significant parts of the Beringian continental shelf
were exposed, cutting off the moisture supply for the Nome River Glaciation (Pushkar et
al., 1999). Subaerial and glaciofluvial deposits above the Nome River tills and correlative
glaciations indicate that Beringian ice retreated, while climate remained cold or grew
colder. Ice wedges and evidence of permafrost are common (Huston et al., 1990; Pushkar
et al., 1999) above Nome River glaciation deposits.
Laminations are again prominent in the sediment record and deposited intermittently
between 394 and 392 ka and again after 375 ka (Fig. 4) as the climate transitioned into
MIS 10. These laminations are quite different from the Termination V Laminations due
to their shorter duration and lack of obvious shift in terrigenous vs. marine carbon source.
In addition, these Type II laminations have increased diatom abundances and $CaCO_3$, but
not necessarily increased upwelling indicators reflecting increased primary production
that is perhaps not linked to nutrient upwelling along the shelf-slope break.
Most of these laminations show an increase in sea ice diatoms and a decrease in
productivity indicators. These roughly correspond with millennial scale stadial events
that occurred during MIS 11a in the North Atlantic (Fig. 7) (Voelker et al., 2010). Late
MIS 11 is characterized as a series of warm and cold cycles (Candy et al., 2014; Voelker
et al., 2010), though there is not agreement on the timing of these cycles.



It is tantalizing to note that the laminations occur at a time when global sea level was fluctuating around -50 mapsl (Rohling et al., 2010) (see grey line at -50 m on Fig. 7). Increased productivity and repeating laminated sediments could be related to shelf to basin nutrient dynamics as rising sea levels carry fresh organic matter from the shelf out over the southern Bering Sea (e.g. Bertrand et al., 2000). In addition to the correspondence between laminations and North Atlantic stadials, carbonate peaks in the Bering Sea also occur coeval with carbonate peaks at Blake Ridge (Chaisson et al., 2002) suggesting teleconnections between productivity in the Bering Sea and the North Atlantic at this time. This suggests that sea level fluctuation driven by the closure of Bering Strait may also be occurring at the end of MIS 11 as well as during the last glacial maximum (Hu et al., 2010), though this hypothesis requires further testing and rethinking of dynamic topography in the Bering Strait region over time.

## 6    Conclusions

The interval between glacial MIS 12 and MIS 10 is marked by large changes in productivity but minor changes in sea ice extent at the shelf slope break in the Bering Sea. There is inconclusive evidence for a reversal of the Bering Strait current at 405 ka, but evidence for teleconnections between the Atlantic and the North Pacific is strong when eustatic sea level fluctuated near the Bering Strait sill depth at the end of MIS 11. Tidewater glaciers advanced in Beringia when eustatic sea level was high, insolation was declining in the Arctic, and other high latitude regions saw decreasing SSTs.

During MIS 12, productivity and nitrogen utilization was low. At Termination V, an 11 kyr long laminated interval began. This interval was highly productive for multiple phytoplankton groups and diatom productivity increased by two orders of magnitude while nitrogen utilization decreased. The surface waters were relatively warm and unstratified with reduced sea ice duration. This period is marked by the highest terrigenous organic matter input of the record possibly due to scouring of the continental shelf as sea level rose. Throughout much of MIS 11, productivity changed in concert with changes in insolation and water temperature. During warmer periods, high stratification



appears to have led to lowered productivity in both the Atlantic and the North Pacific
Oceans.
During MIS 12, seasonal sea ice dominated the western Bering Sea with highly stratified
waters during the ice-melt season. Sea ice was at a minimum from 423 to 410 ka when
the Termination V Laminations were deposited. After this, although summers appear to
have been warm, seasonal sea ice lasted longer. And at the end of MIS 11, sea ice
increased and cooling continued in sync with declining sea level.
While decreased NADW formation and species transport from the Arctic Ocean
southward support a reversal of the Bering Strait current at 405 ka, there is no evidence
for transport of Arctic Ocean clay minerals or an increase in winds in Antarctica. In
addition, oceanographic changes indicative of a shift in Bering Strait through flow occur
in series over 4 kyrs or more and there is no synchronicity with Bering Sea changes,
NADW formation and Antarctic winds. Therefore, there is inconclusive evidence for a
reversal of the Bering Strait current during MIS 11.
When global sea level was at its maximum, insolation dropped slightly and coastal
Beringia began to cool in sync with other polar regions. Tidewater glaciers brought
neritic species far off shore and are attributed to humid conditions in Beringia that
allowed glacial growth. Evidence of glaciation is short lived in the western Bering Sea
and followed by an intensification of northerly winds that brought freshwater diatoms out
over the open ocean.
Laminations at end MIS 11 correspond with millennial scale stadials seen in the N
Atlantic. These deposits represent further possible evidence of teleconnections between
the Atlantic and the Pacific as eustatic sea level fluctuated near the Bering Strait sill
depth.
This study supports hypotheses that the region responds to insolation changes at 65° N
and that Bering Strait modulates climate in both the North Atlantic and Pacific regions.
Future work should focus on leads and lags between changes in the North Atlantic, North
Pacific and Antarctic regions to determine how upwelling, deep water formation, and
climate are related.



Data used in this manuscript are archived at the National Center for Environmental
Information (doi and web address pending).

## Acknowledgements

The authors thank the captain and crew of the JOIDES Resolution and Exp. 323 co-chief
scientists Christina Ravelo and Kozo Takahashi. This work was partially supported by
National Science Foundation, Office of Polar Programs Arctic Natural Sciences Award
#1023537 and a Post Expedition Award from the Consortium for Ocean Leadership.





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





Table 1: Bering Sea diatom species grouped by environmental niche. In cases where a species appears in more than one niche, the
grouping used in this study is highlighted in bold.

| *Modern Seasonal Succession* | | | |
| --- | --- | --- | --- |
| **Epontic** | **Marginal Ice Zone (MIZ)** | **Both Epontic and MIZ** | **Summer Bloom** |
| ***Navicula transitrans*** | ***Bacterosira bathyomphala*** | *Actinocyclus curvatulus* | ***Coscinodiscus* spp.** |
| ***Nitzschia frigida*** | *Chaetoceros furcellatus* | ***Fossula arctica*** | ***Leptocylindrus* sp.** |
| | *Chaetoceros socialis* | ***Fragilariopsis cylindrus*** | ***Rhizosolenia* spp.** |
| | *Leptocylindrus* sp. | ***Fragilariopsis oceanica*** | |
| | *Odontella aurita* | ***Navicula pelagica*** | |
| | *Paralia sulcata* | **Naviculoid pennates** | |
| | ***Porosira glacialis*** | *Nitzschia* **spp.** | |
| | ***Staurosirella* cf. *pinnata*** | ***Pinnularia quadratarea*** | |
| | *Thalassionema nitzschioides* | ***Thalassiosira antarctica*** | |
| | ***Thalassiosira angulata*** | ***Thalassiosira gravida*** | |
| | ***Thalassiosira baltica*** | | |
| | ***Thalassiosira decipiens*** | | |
| | ***Thalassiosira hyalina*** | | |
| | ***Thalassiosira hyperborea*** | | |
| | ***Thalassiosira nordenskioeldii*** | | |
| | *Thalassiosira pacifica* | | |


Continued on next page





1423                                         Table 1 continued

| | Water Mass Tracers | | | Shelf to Basin Transport | |
|---|---|---|---|---|---|
| Dicothermal | High Productivity | Alaska Stream | Warmer Water | Neritic | Fresh Water |
| *Actinocyclus curvatulus* | | *Neodenticula seminae* | | *Actinoptychus senarius* | *Lindavia* **cf.** *ocellata* |
| *Chaetoceros* spp. | | | *Azpeitia tabularis* | | |
| *Thalassiosira trifulta* | | | *Stellarimia stellaris* | | |
| | *Odontella aurita* | | | *Amphora* **sp.** | *Lindavia stylorum* |
| | *Thalassionema nitzschioides* | | *Thalassionema nitzschioides* | | *Staurosirella* cf *pinnata* |
| | *Thalassiosira pacifica* | | *Thalassiosira eccentrica* | *Lindavia stylorum* | |
| | *Thalassiosira* **spp.** **small** | | *Thalassiosira oestrupii* | *Delphineis spp.* | *Lindavia radiosa* |
| | *Thalassiothrix longissima* | | *Thalassiosira symmetrica* | *Dentonula confervacea* | |
| | | | | *Diploneis smithii* | |
| | | | | Naviculoid pennates | |
| | | | | *Odontella aurita* | |
| | | | | *Paralia sulcata* | |
| | | | | *Rhaphoneis amphiceros* | |
| | | | | *Stephanopyxis turris* | |
| | | | | *Thalassiosira angulata* | |
| | | | | *Thalassiosira decipiens* | |
| | | | | *Thalassiosira eccentrica* | |



Table 2: Distribution of Laminated Intervals during MIS 11. Note that the depth and age
1425         of laminated intervals encompasses all holes drilled, but the average duration is
1426         calculated using each of the holes that it is present in.

| Lamination | Type | Depth (mbsf) | Age (ka) | Average Duration (kyrs) | Found in Holes |
|---|---|---|---|---|---|
| MIS 11.5 | II | 112.02-111.47 | 367.2-366.0 | 0.50 | C |
| MIS 11.4 | II | 113.14-112.94 | 369.7-369.3 | 0.34 | CD |
| MIS 11.3 | II | 114.28-113.95 | 372.3-371.5 | 0.73 | D |
| MIS 11.2 | II | 115.59-114.69 | 374.8-373.2 | 1.25 | ACE |
| MIS 11.1 | II | 121.84-121.18 | 394.1-392.1 | 1.10 | ED |
| Termination V | I | 130.00-126.52 | 423.3-410.4 | 12.14 | ACDE |






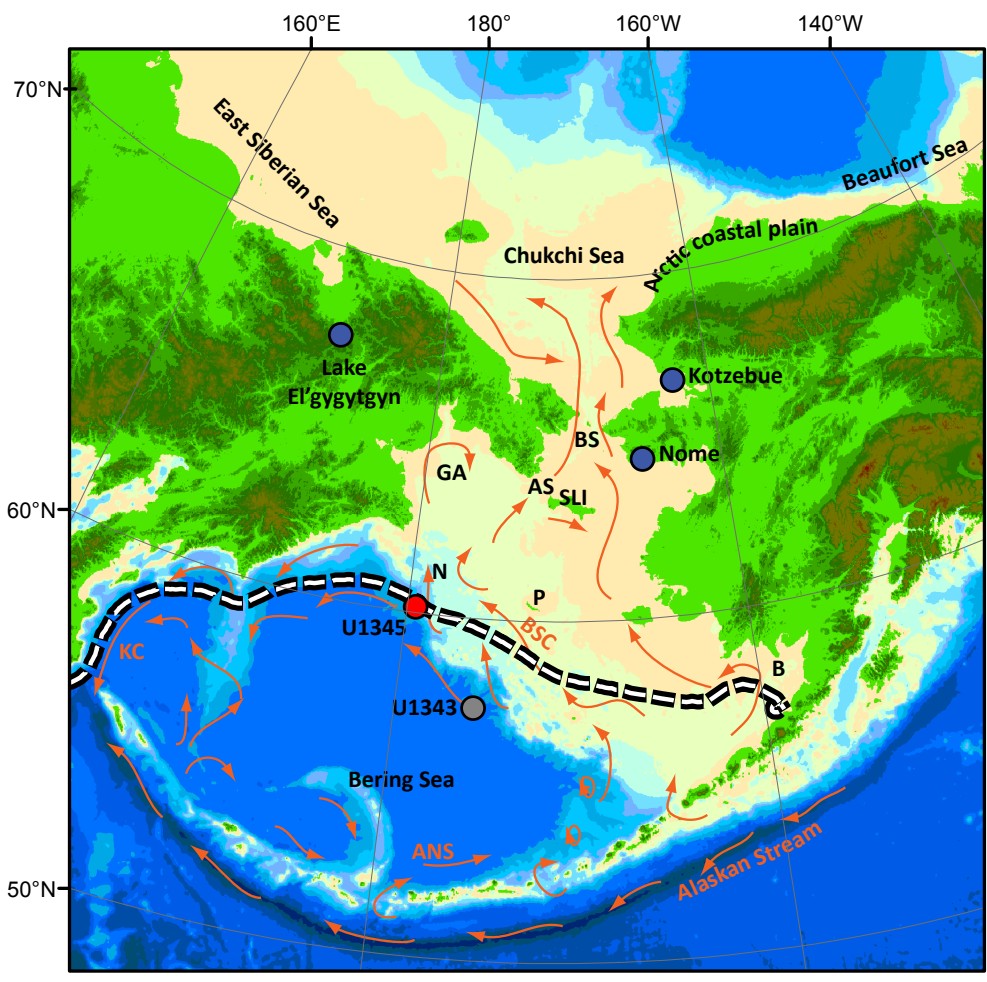

Figure 1. Map of Beringia. Locations of place names from the text are labeled: Aleutian North Slope Current (ANS), Anadyr Strait (AS), Bristol Bay (B), Bering Strait (BS), Bering Slope Current (BSC), Gulf of Anadyr (GA), Kamchatka Current (KC), Navarin Canyon (N), Pribilof Islands (P), St. Lawrence Island (SLI). The white and black dashed line is the modern, median maximum extent of sea ice (Cavalieri et al., 1996). Currents are in orange and are modified from Stabeno (Stabeno et al., 1999). Base map is modified from Manley (Manley, 2002).



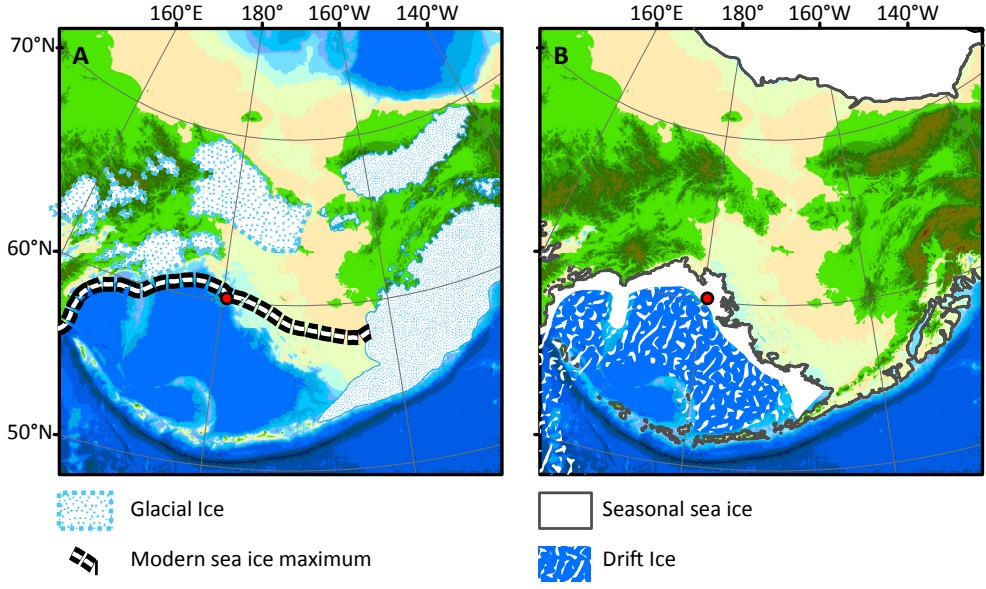

Figure 2. Maximum glacial and sea ice extents in Beringia. A. depicts the maximum glacial ice in Beringia as inferred from terminal and lateral moraines. This is not intended to show the maximum extent during a particular glaciation, but rather the maximum possible extent of glacial ice. These moraines are likely from several different major glaciations. The white and black dashed line is the modern, median maximum extent of sea ice (Cavalieri et al., 1996). B. depicts the approximate pattern of sea ice during glacial stages (Katsuki and Takahashi, 2005). The dark grey contour is -140 m, the approximate sea level during MIS 12 (Rohling et al., 2010). Base map is modified from Manley (Manley, 2002).





Figure 3. Age model. Blue plots depict data from Site U1343, red plots are from U1345.
Magnetic susceptibility and benthic foraminiferal $\delta^{18}O$ are plotted by depth for each Site
in the top half of the figure. The grey line joining the magnetic susceptibility plots
indicates the tie point that was added in this study. Inverted triangles indicate locations of
tie points between Bering Sea $\delta^{18}O$ (Cook et al., In Press; Kim et al., 2014) and the global
marine stack (Lisiecki and Raymo, 2005), which is plotted in grey. Green bars indicate
laminated intervals in U1343 and U1345 plotted by age. The bottom half of the figure
depicts the same data (magnetic susceptibility and $\delta^{18}O$) plotted by age with the global
marine stack (grey) and insolation at 65° N (orange for reference).



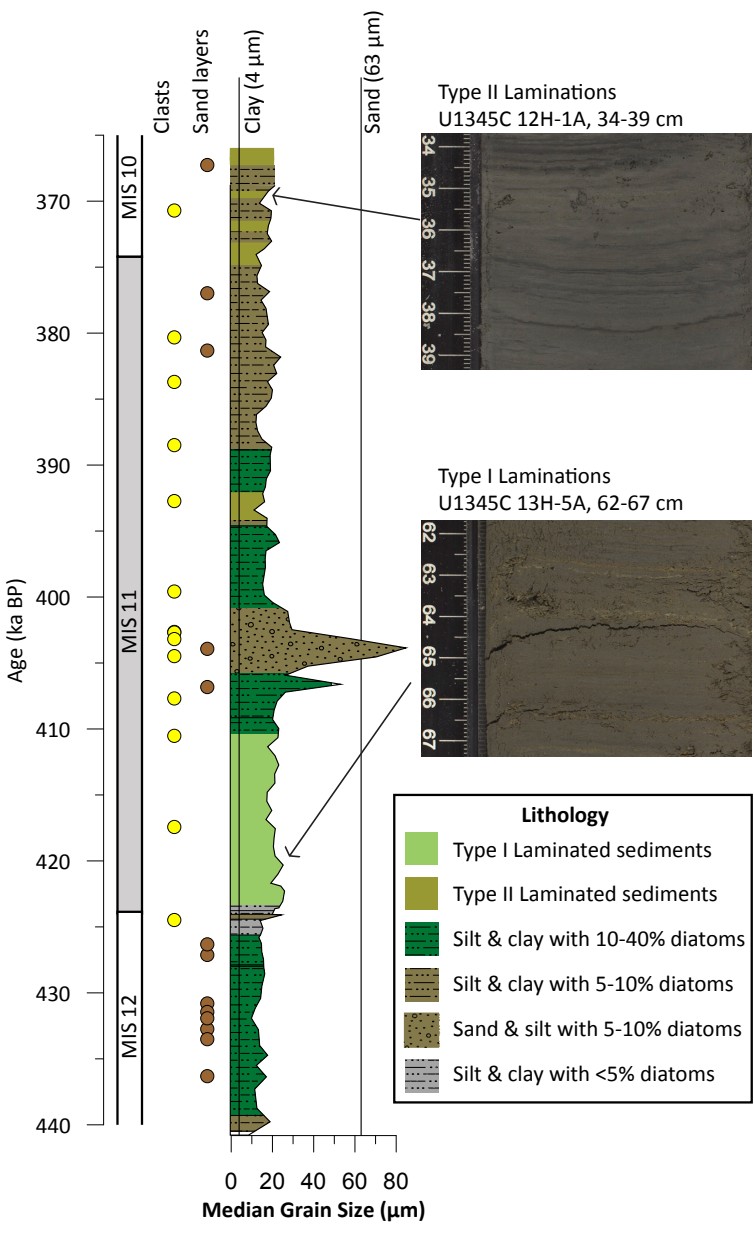

Figure 4. Lithostratigraphic column for U1345A. Marine Isotope Stage 11 is depicted as a grey bar. Ice rafted debris (yellow dots) and sand layers (maroon dots) are a compilation of these features in all four holes at U1345. The width of the lithologic column varies according to median grain size. Vertical lines indicate the cut off for clay and sand sized particles. Silt lies between the two lines. Colors depict varying amounts of diatoms relative to terrigenous grains in the sediment. Type I Laminations are depicted as pale green bars and Type II laminations are depicted as olive green bars. An example of each of the lamination types is shown in the images to the right.

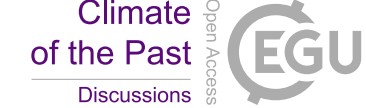

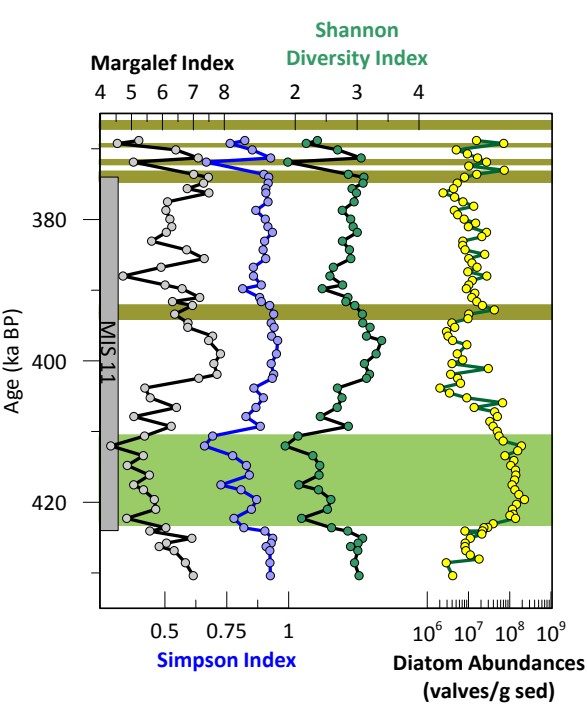

Figure 5. The Margalef, Simpson, and Shannon diversity indices plotted with diatom abundances. Type I Laminations are depicted as pale green bars and Type II laminations are depicted as olive green bars.







Figure 6. Absolute and relative percent abundances of all diatoms that occur in
abundances greater than 10% of any assemblage. Line plots depict absolute abundance
and area plots depict relative percent abundance. Species are color coded according to the
niche that they are grouped into: marginal ice zone (light blue), both ice types (dark blue
to light blue), dicothermal (light green), high productivity (green), neritic (orange),
freshwater (brown), North Pacific (yellow), and warm water (red). Insolation 65° N (light
grey line) is also shown. Type I Laminations are depicted as pale green bars and Type II
laminations are depicted as olive green bars.









Figure 7. Summary of geochemistry and biological proxies. The grey vertical bar depicts the duration of MIS 11, colored vertical bars refer to the sections in the text, and dark grey bars show the timing of North Atlantic stadials (I-IV) (Voelker et al., 2010). Global eustatic sea level (orange) is plotted for reference. The sill depth of Bering Straight (-50 m apsl) is shown as a vertical grey line. Total organic carbon (red) is plotted with the total diatom abundance (green line, yellow dots). Geochemical data is plotted as $\delta^{15}$N (red), C/N (blue), $\delta^{13}$C (black), and % $CaCO_3$ (yellow). Biological proxies include absolute abundance of calcareous nannofossils (purple), and relative percent abundances of diatoms grouped by environmental niche (see color coding in Fig. 6). Insolation at 65° N (black) is overlain on *Chaetoceros* RS relative percent abundances. Type I Laminations are depicted as pale green bars and Type II laminations are depicted as olive green bars. A grey bar indicates the Beringian glacial advance.









Figure 8. Proxy indicators of shelf to basin transport. Total diatom absolute abundances are plotted next to absolute (line plots) and relative percent abundance of *P. sulcata* (orange area plot) and fresh water species (brown area plot). High-resolution grain size includes % clay, sand, and greater than 250 μm (red lines) and % silt and 150-250 μm (black lines). Yellow circles indicate isolated clasts (IRD), maroon circles indicate sand layers for all holes at U1345. Geochemical data is plotted as $\delta^{15}N$ (red), C/N (black), and $\delta^{13}C$ (black). The grey bar spans 406-396 ka, the interval of increased shelf to basin transport in Beringia.