# Peer review of "Bering Sea surface water conditions during Marine Isotope Stages 12 to 10 at Navarin Canyon (IODP Site U1345)"

_Climate of the Past, 2015_

## Referee Comment (RC1) · T. Cronin (Referee) · 11 Feb 2016

This paper reconstructs paleoceanography for an important glacial-interglacial cycle, MIS 12-11 in an important region in the Bering Sea using IODP cores. MIS 11 is an especially important interglacial due to pre industrial-level $CO_2$ concentrations but higher-than-present sea level and in many regions, significantly warmer air and sea temperatures. The study uses sediment, geochemical and micropaleo proxies (diatoms & calcareous nannofossils) for marine productivity, sea ice and land ice reconstruction. So I think a high-latitude marine record of the MIS12-11 period like this one is sorely needed to go with Lake E, Lake B and others.

Some general issues I think the authors should deal with in a revision:
[Figure]

1) the introduction tends to be unfocused and too long. Please shorten it giving the main hypotheses to be tested. I also think the methods and results sections tend to be long. In section 4.2.2, in the diatom ecology section, can the main taxa or assemblages used for productivity and sea ice be emphasized? Much of this section is taken care of in the table. In fact, it really constitutes a review of high latitude diatom ecology going back to Sancetta, Koizuma and other pioneers; this is useful but it merits its own paper in a micropaleo journal.

2) The methods section is long and could be put in a supplement, in fact some of it is, but the Supplementary Materials is not cited until Page 28.

Likewise, Section 5 is too descriptive and does not focus on the key patterns that address the hypothesis about suborbital variability.

3) related to # 1, I sense some of the text and references are not quite up to date [interglacial sea level papers, Bering Sea sea-level section 2.2 [where is Keigwin 2006 paper?), modern Bering Sea oceanography, see section 2.3)

4) The NADW discussion does not belong under a section called Bering Sea hydrology. Later in the paper, Section 5, there is again NADW discussion in the context of late MIS 11 Bering Sea reversed flow. In general, I don't think the Bering-N Atlantic links are well established mainly due to chronology/correlation issues, which I believe are discussed in a 2009 paper by L&R on Atlantic Pacific diachroneity of O18 records.

5) I wonder if the suggested correlations of this study's IODP core records with emerged Quaternary marine deposits [this is mentioned in several places} are warranted given age uncertainty of the onshore deposits?

6) The 404 ka ice-rafting event discussed on page 28 seems speculative and not up to date on ice-rafting processes in the Arctic and subarctic. This section evolves into a mechanistic explanation, covering the "snow gun" hypothesis and alternatives [turbidites, sea ice etc]. I think this section should be rethought and rewritten. As with the

issue of Bering Sea flow reversal in an earlier section, are these central to the question of patterns and causes of variability within MIS11?

7) The paper uses both cores - U1345 & 1343 - although Kim published on U1343 using different proxies but the same O18 for tuning, is there any way to integrate results from both cores better to provide a more robust pattern of MIS paleoceanography?

Is the main focus of the study on orbital glacial-interglacial timescales or millennial timescales (that is, stadials and interstadials within MIS 11, see section 2.1 on sea level, or abrupt reversals like DO events ? The 15-meter thick MIS 11 record [line 199] ought to allow millennial-scale events to be seen. I have concern with the authors statement, in their discussion of the age model and tuning to LR04 and the other site U1343: "we urge caution when interpreting millennial scale changes at the site or comparing our record to others that examine MIS 11 at millennial scale resolution or finer". I got the impression in the introduction there would be more definitive conclusions reached on within-interglacial climate variability. Plots in Figures 5-8 don't really show me DO-like or Heinrich-like variability, which could be an important new conclusion, given our ideas on what causes such events in at least the N Atlantic region.

In sum, I rated the paper as accept after minor revisions, some changes I am suggesting might take major text-shortening, but the science presented is sound, it is just not clearly packaged or presented.

Specific Comments

Line 27 comma before however

Line 28: This confuses me as the paper is in the Bering Sea, not the N Atlantic: led to "lowered productivity in both the northern Atlantic and the northern Pacific. "

Line 48 proper citation of IPCC 2013

Lines 55-56. Fix grammar in second part of sentence.

Line 58 – which coastal region were these glaciers?

Line 70 – do you mean little is known from North Pacific Ocean region incl. Bering Sea?

Line 74. Is this marginal zone sea ice?

Line 114 E Antarctic ice was stable. . .

Line 196 "and" no italics

Section 4.2.1. Authors begin to use "ka" in discussions of diatoms but absolute years were not discussed in the age model-tuning section. So please tell readers earlier in the paper, at Table 2 reference, which should be in age model section, about the ages of MIS12, MIS 11 per the tuning to LR04. See line 586- the section title should say how old oldest sediments are, not "beginning of record"

Section 5.1 includes early deglaciation but section 5.2 is on Termination V, which is the deglaciation. Line 625, the debate about the duration of MIS 11 should be mentioned, embodied in papers by Masson-Delmotte, Ruddiman and others. See line 710 on this topic. I became confused about the MIS11 duration and the number of substages. Many authors do NOT use the substage terminology and the LR04 and the Antarctic ice core records show only two MIS 11 peak warm periods. One reason this is critical is this study of MIS 11 in the Bering Sea is one of the most detailed available. So it should shed light on the issue.

Figure 1 in the caption, mention both U-cores plotted in the map.

Figure 3 is a little complicated but it is critical. Consider dividing into 2 figures. The U1345 curve, red line, certainly looks different from that for U1343 – why so? Cook and Kim age model papers might be summarized in the text, in fact it would be useful to reproduce the O18, tie points data for the entire period covered by their tuning study.

Figures 5-8 are fine and do show what the study sought to accomplish: variability in

diversity, lithology and microfossil assemblages. A more succinct treatment of these data in the text and better summary in the conclusions would help readers not familiar with these proxies. Why is MIS 11 split on the left in Fig 7 but not in others? Label horizontal colored panels in the figures for clarity. Figure 7 what is the source of N Atlantic stadials? Are they really relevant to this study?

―――――――――――――――――

---

## Referee Comment (RC2) · J. Addison (Referee) · 22 Mar 2016

The paper presented by Caissie and her colleagues details the development of oceano-graphic conditions in the Bering Sea prior, during, and immediately following Marine Isotope Stage 11. The focus on MIS 11 is timely due to its environmental similarities to predicted near-future climate change, and the Bering Sea geography provides an environment that is of broad interest across many disciplines. This study is also one of only a handful that documents marine environmental change in the high-latitudes during MIS 11, which adds to this study's timeliness. The authors use a combination of diatom and calcareous nannofossil micropaleontology, bulk sediment geochemistry, and grain-size analyses to show the evolution of this interesting period though changes

in the marine ecosystem, sea ice conditions, and water column nutrient cycling.

There are several elements of this study that are great, including detailed environmental changes associated with the various phases of the MIS 12-10 transition, and an honest assessment of the age control. I also particularly liked seeing the application of modern species richness indices to the diatom data in this paper.

Some issues the authors need to address include:

1) Better integration of these new Exp. 323 results with the other recent papers that have resulted from the cruise [e.g., d15N studies of Schlung et al. (2013) and Knudson & Ravelo (2015); opal productivity studies of Kanematsu et al. (2013) and Kim et al. (2014)]. While these studies do not provide as detailed an analysis of MIS 11 as the current paper, they do provide a good background for assessing glacial/interglacial background changes in the Bering Sea that are relevant to the current study.

2) To better assess the relative contributions of terrigenous versus marine organic matter to the dataset, cross-plots of the organic matter d13C, sedimentary d15N, and molar N/C ratios (see Perdue and Koprivnjak (2007) for explanation of N/C instead of C/N for % terrestrial calculations) need to be presented. See Walinsky et al. (2009)'s Figure 9 for a good example. It might also be worth considering breaking the data into groups based on the time intervals introduced in the discussion.

3) During the time periods associated with low sea-level stands in this paper, the mouths of the Yukon and Kuskokwim Rivers (and other smaller rivers that currently drain into the Bering Sea) would have been greatly advanced across the exposed shallow continental shelf. Are these the "glacial meltwater rivers" that are suggested in Section 5.1? It is difficult to dismiss them as potential sources of terrigenous material, especially given the evidence that they contributed an enormous sediment load to the glacial Bering Sea [as evinced at the Meiji Drift, see VanLaningham et al., (2009)], as well as cut some of the largest submarine canyons in the world during these low stands (e.g., Scholl et al. 1970 and subsequent work). Additional explanation for why

Site U1345 appears to be devoid of this terrigenous material seems warranted.

4) As written, the entire Discussion section is tough to follow. There are quite a few time overlaps between the various subsections that are confusing, plus the added details from the contemporaneous North Atlantic and Antarctic regions add further complexities. I recommend re-organizing the Discussion into 2 major sections – (1) the MIS 12-10 transitions as seen at U1345 [subsections for each time interval (without time overlaps), which is similar to what has already been written], and relating the U1345 variability to other regional/global records.

5) Since the original premise of this study was intended to present the Bering Sea MIS 11 paleoceanographic variability as an analogue for future conditions, perhaps a small section at the end of the discussion should address this?

6) I'm skeptical about the nature of the deposit that is attributed to being evidence of the Bering Strait Current Reversal (Subsection 5.3.1). When I first saw the grain-size data, I thought turbidite, and the enrichment in P. sulcata [a common diatom marker of redeposition and/or downslope transport due to its highly silicified morphology; see Sancetta (1982)] seems to support that idea. However, the authors discount the turbidite mechanism on account of no visible sedimentary structures that are normally associated with turbidites. However, the authors make a good point about illite being an additional potential Arctic Ocean flow marker (Lines 767-771), as well as being a potential way to explain the anomalous N data. I highly recommend the authors do a little XRD analysis on the sediments in this interval (and immediately preceeding/succeeding) to determine presence/absence of illite in this interval. It is pretty easy, and the lead author's institute has an appropriate instrument (housed in ISU's Office of Biotechnology; www.marl.iastate.edu/xrd.html). This will serve as both an additional line of evidence to support the idea of an Arctic Ocean inflow, as well as help to explain the N data (since the low d15N values suggest an increase in the relative proportion of terrigenous organic matter, not necessarily inorganic N hosted in clays).

7) The idea that the Nome River Glaciation started during peak warmth in MIS 11 is a bit counter-intuitive; I think a better treatment of the extant Nome River Glaciation sites (and in particular, their respective age controls) is required to support this idea. Also, while the authors do introduce the "snow-gun" hypothesis near the end of Subsection 5.3.2, I think re-organization to increase clarity and introduce the snow-gun idea sooner will greatly improve the readability here.

There are a few minor issues as well: 1) Overall, the mean d15N = 6.4‰ for the full dataset, and from looking at Fig. 7, it looks like there might be values that exceed 8 or 9‰These high values are suggestive of denitrification, yet this process isn't considered in the N cycle discussions spread throughout the paper.

2) Because many of the figures are very data-rich, in many cases axes have been truncated, which makes it difficult to assess extreme data points (which are often very important, such as the extremely low d15N values associated with the 406-402 ka event). I would recommend that, instead of cutting axes ranges, they should instead be offset so that the full axis range can be indicated. I'm specifically thinking of Figure 7, but this could apply to many other figures, too. There are also several instances where it is difficult to determine which line goes with which axis; perhaps color coding or additional labels are necessary.

3) I am also providing a PDF copy of the manuscript that I have made several grammatical corrections to; please review in detail.

In conclusion, I would like to recommend this article for acceptance, pending the minor revisions I've indicated here, as well as the editorial revisions on the attached manuscript. If any of my notes are not clear (or legible), I recommend the authors contact me directly with any questions they may have.

Sincerely, Jason A. Addison, PhD US Geological Survey jaddison@usgs.gov

Please also note the supplement to this comment:

[Figure]

http://www.clim-past-discuss.net/cp-2015-184/cp-2015-184-RC2-supplement.pdf

[Figure]

**Supplement:**

[revised manuscript text omitted]

*Isotope standards?*
*(Please include "standard" isotope details,*
*e.g., referenced to which int. standards,*
*‰ notation, etc.)*

**4    Results**

**4.1 Sedimentology**

In general the sediments at Site U1345 are massive with centimeter–scale dark or coarse-grained mottles. The sediments are mainly composed of clay and silt with varying amounts of diatoms, sand, and tephra throughout. Laminated intervals bracket MIS 11 (Fig. 4). The proportion of diatoms relative to terrigenous or volcanogenic grains is highest during laminated intervals and lowest immediately preceding Termination V (~425 ka). Vesiculated tephra shards were seen in every diatom slide analyzed. Several thin (< 1 cm) sand layers and shell fragments were visible on the split cores, especially during MIS 12. However, high-resolution grain size analyses show that the median grain size was lowest during MIS 12, increasing from approximately 14 μm to 21 μm at the start of Termination V at 424.5 ka (130.92 mbsf). Median grain size peaks at 84 μm between 401 and 407 ka (125.42-123.62 mbsf). This interval is also the location of an obvious sandy layer in the core. After this interval, median grain size remains steady at about 17 μm. Subrounded to rounded clasts (granule to pebble) commonly occur on the split surface of the cores. We combined clast and sand layer data from all Holes at Site U1345 when examining their distribution (Fig. 4).

*Is this a turbidite? If so, then "continuous" diagnosis problematic → low diatom abundance → low $\delta^{15}N$ → high P. → sulfate?*

*Transported*

[revised manuscript text omitted]

Barron '09 classifies P. sulcata as "transported" in the GoAK + see complications raised by Ren '14

[Figure]

**4.4 Geochemistry *[handwritten: → missing references to figures]**

**4.4.1 Organic and Inorganic Carbon Content**

Total organic carbon (TOC) roughly follows the trend of relative percent abundances of *Chaetoceros* RS, with higher values during the Termination V Laminations. Mean TOC value during MIS 12 is 0.76%, and during the Termination V Laminations, it is 1.11%. TOC decreases temporarily in sync with depleted $\delta^{15}N$ values, before rising linearly from 404 ka (124.77 mbsf) to 374 ka (115.39 mbsf). TOC is again high during the late MIS 11/MIS 10 laminations.

In contrast, inorganic carbon, calculated as % $CaCO_3$ is less than 1% for most of the record; however, it increases up to 3.5% during the laminated intervals and also at 382 ka (117.87 mbsf), 392 ka (110.00 mbsf), and 408 ka (125.82 mbsf).

**4.4.2 Terrigenous Input Indicator (C/N) *[handwritten: → Molar or atomic?]**

The ratio, C/N is one of two proxies used as indicators of marine versus terrigenous organic matter, with marine values typically ranging from 5-7 and terrigenous ratios over 20 (Meyers, 1994; Redfield et al., 1963).

Throughout the record, C/N indicates primarily a marine source for organic matter. During MIS 12, C/N is highly variable, when sea level is below -50 m apsl. As sea level rises during Termination V, C/N values increase from 6 to more than 9. The highest C/N value occurs at the start of the Termination V Laminations. C/N decreases as sea level rises until at 400 ka (123.62 mbsf) it stabilizes near 7 for the remainder of the record.

**4.4.3 Bulk Sedimentary Stable Isotopes *[handwritten: → Section too brief for those complex data!]**

**4.4.3.1 Carbon Isotopes**

Stable isotopes of carbon are also used as an indicator of marine vs. terrigenous organic matter with $\delta^{13}C$ values near -27 indicating C3 plant-sourced organic matter; values between -22 and -19 are typical for Arctic Ocean marine phytoplankton and -18.3 is average for ice-related plankton (Schubert and Calvert, 2001). However, it has been

*[handwritten margin note, left: Need cross-plots of $\delta^{13}C$, N/C, + $\delta^{15}N$ for the OM source discussion]*

*[handwritten margin note, right: Both Meyers + Redfield use molar, but EA output is commonly atomic]*

[Figure]

*[handwritten: Ref; see (at least) Rao et al. '89 + Laws et al. '95]*

*[handwritten: Statistics?]*

shown that δ$^{13}$C is sometimes related more to growth rate, cell size, and cell membrane permeability, so it may reflect changing phytoplankton groups instead of simply marine vs. C3 plant sources of organic matter in U1345.

Carbon isotopic values range between -22 ‰ and -26 ‰ and are generally anticorrelated with C/N values. These values indicate a mix of marine phytoplankton and C3 plants as the main contributors to organic matter at the site. At the onset of the Termination V

Laminations, δ$^{13}$C becomes more negative and then gradually increases to a maximum of

-22.33 at 404 ka (124.62). After 400 ka (123.5 mbsf), δ$^{13}$C is relatively stable around -

23.5‰.

### 4.4.3.2     Nitrogen Isotopes

Nitrogen, in the form of nitrate, is a key nutrient for phytoplankton growth. Diatoms preferentially assimilate the lighter isotope, $^{14}$N, which in turn enriches surface waters with respect to $^{15}$N (Barron et al., 2009; Shiga and Koizumi, 2000). Keeping in mind the effects of nitrification of oxygen rich and poor sediments (Brunelle et al., 2007), the efficiency of nitrogen utilization can be estimated by examining the $^{15}$N/$^{14}$N ratio of nitrogen in either bulk sedimentary organic matter, with enriched values of δ$^{15}$N

indicating higher nutrient utilization. Sponge spicules (very low δ$^{15}$N values) and radiolarians (highly variable δ$^{15}$N values) may contaminate the δ$^{15}$N of bulk organic matter, however we looked for and found no correlation between spicule abundance and

δ$^{15}$N in our samples.

*[handwritten: No δ$^{15}$N (or N!) in this at all! Maybe you mean Addison '12? in Paleoceanography]*

Surprisingly, δ$^{15}$N is relatively stable throughout the study interval, fluctuating around an average value of 6.4‰, though there are several notable excursions. Coeval with sea level rise and increased relative percent *Chaetoceros* RS, δ$^{15}$N decreased 2.7‰ to 4.4‰ before recovering to average values during the Termination V Laminations. Two other depletions occur at 405 ka (124.77 mbsf) and 393 ka (121.62 mbsf), the first is the most extreme and reaches 2.9‰.

*[handwritten: This is fairly high, could there be an influence from denitrification?]*

[Figure]

**5   Discussion**

[revised manuscript text omitted]

---

## Author Response (AR1)

**Reviewer 1 (T. Cronin):**

**Thank you for this very helpful review. Following these suggestions, we are working on a much shorter manuscript, which includes pulling out a section to develop a separate publication. The most significant revisions that you will notice include streamlining the hypotheses (1. Orbital scale variability; 2. Millenial-scale variability; 3. Direction of Bering Strait throughflow) and better focusing the paper.**

**Below we have included specific responses to the general issues that Dr. Cronin raised. Please note that all minor, line-by-line suggestions will be completed in the final paper.**

This paper reconstructs paleoceanography for an important glacial-interglacial cycle, MIS 12-11 in an important region in the Bering Sea using IODP cores. MIS 11 is an especially important interglacial due to pre industrial-level $CO_2$ concentrations but higher-than-present sea level and in many regions, significantly warmer air and sea temperatures. The study uses sediment, geochemical and micropaleo proxies (diatoms & calcareous nannofossils) for marine productivity, sea ice and land ice reconstruction. So I think a high-latitude marine record of the MIS12-11 period like this one is sorely needed to go with Lake E, Lake B and others.

Some general issues I think the authors should deal with in a revision:

1) the introduction tends to be unfocused and too long. Please shorten it giving the main hypotheses to be tested. I also think the methods and results sections tend to be long. In section 4.2.2, in the diatom ecology section, can the main taxa or assemblages used for productivity and sea ice be emphasized? Much of this section is taken care of in the table. In fact, it really constitutes a review of high latitude diatom ecology going back to Sancetta, Koizuma and other pioneers; this is useful but it merits its own paper in a micropaleo journal.

**Thank you for this fabulous suggestion. The diatom ecology section has been removed from this paper and a separate manuscript is nearly ready to be submitted based on this (and other relevant) work. Additionally, the introduction, methods and results have all been shortened quite a bit.**

2) The methods section is long and could be put in a supplement, in fact some of it is, but the Supplementary Materials is not cited until Page 28. Likewise, Section 5 is too descriptive and does not focus on the key patterns that address the hypothesis about suborbital variability.

**Much of the methods section has been pulled out of the main text and put into a new methods supplement. This supplement also contains additional information requested by the second reviewer.**

3) related to # 1, I sense some of the text and references are not quite up to date [interglacial sea level papers, Bering Sea sea-level section 2.2 [where is Keigwin 2006 paper?), modern Bering Sea oceanography, see section 2.3)

**JULIE:**
**Section 2.2 intentionally left out the Keigwin paper because it was intended to be more general than the most recent deglaciation and also focused more on terrigenous input rather than timing of sea level rise, however, the Keigwin paper is important, so we've added it. We will of course also make reference to the sea level compilation of Kaufman and Brigham-Grette, 1993 in the context of the compilation by Rohling et al. 2014 in Nature. In addition, the following papers, particularly Bering Sea interglacial papers that were not yet in press when we first submitted this paper, have also been added: $\delta^{15}N$ studies (Schlung et al., 2013; Knudson and Ravelo, 2015); opal productivity (Kanematsu et al., 2013; Kim et al., 2013); clay mineralogy (Kim, 2015), and the diatom study from Teraishi (2015).**

4) The NADW discussion does not belong under a section called Bering Sea hydrology. Later in the paper, Section 5, there is again NADW discussion in the context of late MIS 11 Bering Sea reversed flow. In general, I don't think the Bering-N Atlantic links are well established mainly due to chronology/correlation issues, which I believe are discussed in a 2009 paper by L&R on Atlantic Pacific diachroneity of O18 records.

**Discussion of NADW was included in this section because of its hypothesized influence on the direction of flow through Bering Strait, however we agree that this discussion is misplaced and you are correct that it should not be possible to determine millennial-scale synchronicity between the Atlantic and Pacific. The association with NADW has been removed from the paper and the issue of the direction of throughflow is included later in the paper when this hypothesis is tested.**

5) I wonder if the suggested correlations of this study's IODP core records with emerged Quaternary marine deposits [this is mentioned in several places] are warranted given age uncertainty of the onshore deposits?

**The chronology of the onshore deposits is certainly less accurate that a marine core will ever be. But we would like to argue that Kaufman and Brigham-Grette, 1993 and Pushkar et al., 1999 provide the most likely interglacial age for the Nome River Glaciation. The stratigraphy there places important constraints on early ice build up in local mountain ranges before global sea level drops. Kaufman et al., 2001 have the same advance in the Bristol Bay region. Our findings can add support to the onshore chronology.**

6) The 404 ka ice-rafting event discussed on page 28 seems speculative and not up to date on ice-rafting processes in the Arctic and subarctic. This section evolves into a mechanistic explanation, covering the "snow gun" hypothesis and alternatives [turbidites, sea ice etc]. I think this section should be rethought and rewritten. As with the issue of Bering Sea flow reversal in an earlier section, are these central to the question of patterns and causes of variability within MIS11?

**This section has been significantly updated and simplified to include the comments of Reviewer 2, who asked us to more fully explore the question of a turbidite during this interval. While we agree that the hypothesized glacial advance is likely not adequately tested, there are advances during MIS 11 and 5e/5d transition (the latter not important here), which do provide an important means of linking land and sea responses. This is something the Arctic Ocean records cannot do. We suggest that we reframe the discussion about this aspect into a speculative section that could drive new work to explore the sources of the IRD.**

7) The paper uses both cores - U1345 & 1343 - although Kim published on U1343 using different proxies but the same O18 for tuning, is there any way to integrate results from both cores better to provide a more robust pattern of MIS paleoceanography?

**This is an excellent point. Kim's 2013 and 2015 low resolution opal and clay mineralogy papers will be incorporated.**

Is the main focus of the study on orbital glacial-interglacial timescales or millennial timescales (that is, stadials and interstadials within MIS 11, see section 2.1 on sea level, or abrupt reversals like DO events ? The 15-meter thick MIS 11 record [line 199] ought to allow millennial-scale events to be seen. I have concern with the authors statement, in their discussion of the age model and tuning to LR04 and the other site U1343: "we urge caution when interpreting millennial scale changes at the site or comparing our record to others that examine MIS 11 at millennial scale resolution or finer". I got the impression in the introduction there would be more definitive conclusions reached on within-interglacial climate variability. Plots in Figures 5-8 don't really show me DOlike or Heinrich-like variability, which could be an important new conclusion, given our ideas on what causes such events in at least the N Atlantic region.

**There are 3 main hypotheses that this paper seeks to test:**
**1. Productivity and sea ice extent are primarily controlled by orbital-scale forcing MIS 11**
**2. Millennial-scale changes in sea ice occur throughout MIS 11**
**3. Throughflow through Bering Strait temporarily reversed after Termination V**

**Additionally, we speculate that continuous marine records in the Bering Sea include records of glacial advance that can be used to explore land-sea linkages, but this section is significantly shortened and not treated as equal to the above three hypotheses.**

**We understand your concern about our caution about age model error, however we think it is important to recognize the limitations of the age model, especially in light of questions raised by Liesiecki and Raymo (2009) about synchronicity (or lack there of) of the isotope stack between the North Atlantic and Pacific. We think that the age model allows a reasonable estimate of sedimentation rates in the core, and the ages are likely fairly precise, however, the error in LR04 is 4 kyrs for sediments younger than 1 Ma. This means that it is not possible to say that an event that happened in U1345 at 412.4 corresponds with an event at 412.4 in a distal core. However, we can certainly resolve events at millennial scales within this core. Perhaps it makes the most sense to keep the caution about interpreting millennial-scale events BETWEEN cores, but remove the line about interpreting millennial scale changes within U1345.**

In sum, I rated the paper as accept after minor revisions, some changes I am suggesting might take major text-shortening, but the science presented is sound, it is just not clearly packaged or presented.

Specific Comments

Line 27 comma before however
Line 28: This confuses me as the paper is in the Bering Sea, not the N Atlantic: led to "lowered productivity in both the northern Atlantic and the northern Pacific. "
Line 48 proper citation of IPCC 2013
Lines 55-56. Fix grammar in second part of sentence. **This sentence was cut.**
Line 58 – which coastal region were these glaciers? **This sentence was cut.**
Line 70 – do you mean little is known from North Pacific Ocean region incl. Bering Sea?
Line 74. Is this marginal zone sea ice?
Line 114 E Antarctic ice was stable. . .
Line 196 "and" no italics

Section 4.2.1. Authors begin to use "ka" in discussions of diatoms but absolute years were not discussed in the age model-tuning section. So please tell readers earlier in the paper, at Table 2 reference, which should be in age model section, about the ages of MIS12, MIS 11 per the tuning to LR04. See line 586- the section title should say how old oldest sediments are, not "beginning of record"

**We agree with these changes.**

Section 5.1 includes early deglaciation but section 5.2 is on Termination V, which is the deglaciation. Line 625, the debate about the duration of MIS 11 should be mentioned, embodied in papers by Masson-Delmotte, Ruddiman and others. See line 710 on this topic. I became confused about the MIS11 duration and the number of substages. Many authors do NOT use the substage terminology and the LR04 and the Antarctic ice core records show only two MIS 11 peak warm periods. One reason this is critical is this study of MIS 11 in the Bering Sea is one of the most detailed available. So it should shed light on the issue.

**We will address this issue in the revisions and appreciate this important point.**

Figure 1 in the caption, mention both U-cores plotted in the map.

Figure 3 is a little complicated but it is critical. Consider dividing into 2 figures. The U1345 curve, red line, certainly looks different from that for U1343 – why so? Cook and Kim age model papers might be summarized in the text, in fact it would be useful to reproduce the O18, tie points data for the entire period covered by their tuning study.

**A table will be added.**

Figures 5-8 are fine and do show what the study sought to accomplish: variability in diversity, lithology and microfossil assemblages. A more succinct treatment of these data in the text and better summary in the conclusions would help readers not familiar with these proxies. Why is MIS 11 split on the left in Fig 7 but not in others? Label horizontal colored panels in the figures for clarity. Figure 7 what is the source of N Atlantic stadials? Are they really relevant to this study?

**We will correct the figures.**

**Reviewer 2: J. Addison**

The paper presented by Caissie and her colleagues details the development of oceanographic conditions in the Bering Sea prior, during, and immediately following Marine Isotope Stage 11. The focus on MIS 11 is timely due to its environmental similarities to predicted near-future climate change, and the Bering Sea geography provides an environment that is of broad interest across many disciplines. This study is also one of only a handful that documents marine environmental change in the high-latitudes during MIS 11, which adds to this study's timeliness. The authors use a combination of diatom and calcareous nannofossil micropaleontology, bulk sediment geochemistry, and grain-size analyses to show the evolution of this interesting period though changes in the marine ecosystem, sea ice conditions, and water column nutrient cycling. There are several elements of this study that are great, including detailed environmental changes associated with the various phases of the MIS 12-10 transition, and an honest assessment of the age control. I also particularly liked seeing the application of modern species richness indices to the diatom data in this paper.

Some issues the authors need to address include:

> 1) Better integration of these new Exp. 323 results with the other recent papers that have resulted from the cruise [e.g., d15N studies of Schlung et al. (2013) and Knudson & Ravelo (2015); opal productivity studies of Kanematsu et al. (2013) and Kim et al. (2014)]. While these studies do not provide as detailed an analysis of MIS 11 as the current paper, they do provide a good background for assessing glacial/interglacial background changes in the Bering Sea that are relevant to the current study.

**These records will be assessed and integrated into this manuscript. Thank you for pointing out their omission.**

> 2) To better assess the relative contributions of terrigenous versus marine organic matter to the dataset, cross-plots of the organic matter d13C, sedimentary d15N, and molar N/C ratios (see Perdue and Koprivnjak (2007) for explanation of N/C instead of C/N for % terrestrial calculations) need to be presented. See Walinsky et al. (2009)'s Figure 9 for a good example. It might also be worth considering breaking the data into groups based on the time intervals introduced in the discussion.

**Thank you for this suggestion. This will help frame the discussion about contribution of terrigenous matter and organic matter as well as help us interpret sub-millennial scale variability.**

> 3) During the time periods associated with low sea-level stands in this paper, the mouths of the Yukon and Kuskokwim Rivers (and other smaller rivers that currently drain into the Bering Sea) would have been greatly advanced across the exposed shallow continental shelf. Are these the "glacial meltwater rivers" that are suggested in Section 5.1? It is difficult to dismiss them as potential sources of terrigenous material, especially given the evidence that they contributed an enormous sediment load to the glacial Bering Sea [as evinced at the Meiji Drift, see VanLaningham et al., (2009)], as well as cut some of the largest submarine canyons in the world during these low stands (e.g., Scholl et al. 1970 and subsequent work). Additional explanation for why Site U1345 appears to be devoid of this terrigenous material seems warranted.

**This is an excellent point, we did not intend to dismiss the sedimentation from these rivers. In light of the new clay mineralogy data (see below), this interpretation has been revisited. Pelto et al, in revision, shows a decrease in the input of sediment to a nearby site from the Yukon as sea level rose during the Deglaciation. Our work on this older time period (MIS 11) can reference Pelto (in revision) for additional context.**

> 4) As written, the entire Discussion section is tough to follow. There are quite a few time overlaps between the various subsections that are confusing, plus the added details from the contemporaneous North Atlantic and Antarctic regions add further complexities. I recommend re-organizing the Discussion into 2 major sections – (1) the MIS 12-10 transitions as seen at U1345 [subsections for each time interval (without time overlaps), which is similar to what has already been written], and relating the U1345 variability to other regional/global records.

**This suggestion, combined with the suggestions of reviewer 1 should make the discussion shorter and much more readable.**

> 5) Since the original premise of this study was intended to present the Bering Sea MIS 11 paleoceanographic variability as an analogue for future conditions, perhaps a small section at the end of the discussion should address this?

**This will be added to the final paper in addition to a short summary of the question about the length of MIS 11.**

> 6) I'm skeptical about the nature of the deposit that is attributed to being evidence of the Bering Strait Current Reversal (Subsection 5.3.1). When I first saw the grain-size data, I thought turbidite, and the enrichment in P. sulcata [a common diatom marker of redeposition and/or downslope transport due to its highly silicified morphology; see Sancetta (1982)] seems to support that idea. However, the authors discount the turbidite mechanism on account of no visible sedimentary structures that are normally associated with turbidites. However, the authors make a good point about illite being an additional potential Arctic Ocean flow marker (Lines 767-771), as well as being a potential way to explain the anomalous N data. I highly recommend the authors do a little XRD analysis on the sediments in this interval (and immediately preceeding/succeeding) to determine presence/absence of illite in this interval. It is pretty easy, and the lead author's institute has an appropriate instrument (housed in ISU's Office of Biotechnology; www.marl.iastate.edu/xrd.html). This will serve as both an additional line of evidence to support the idea of an Arctic Ocean inflow, as well as help to explain the N data (since the low d15N values suggest an increase in the relative proportion of terrigenous organic matter, not necessarily inorganic N hosted in clays).

**Thank you for asking us to take a closer look at this interval. The nature of this anomalous deposit remains unknown, though a turbidite is certainly possible. However, the site's position was chosen on an interfluve to avoid turbidites as much as possible. Moreover, the typical graded layers, from coarse sand and microconglomerates in the bottom to silt and clay in the top (Bouma sequence) are missing from this site. Instead we see very poorly sorted terrigenous fragments mixed together. It's unusual to get just one layer like this. We would expect a turbidite to work in the same place for a prolonged period. If the low $\delta^{15}N$ suggested an increase in terrigenous organic matter, we would expect to see a change in C/N and/or $\delta^{13}C$ as well. There is nothing remarkable about either marker during these low $\delta^{15}N$ excursions.**

**In addition, as suggested, we have analyzed the clay mineralogy in 10 samples across MIS 11, including several in the proposed glacial advance/throughflow reversal interval and found no evidence of illite in the core, so an Arctic Ocean influence is unlikely at U1345. This is in direct contrast to the results of Kim et al., 2015 who saw large amounts of illite nearby in U1343. It may be that the currents were such that the same events are not recorded everywhere, though this seems unlikely. Our revision examines these two interpretations and addresses possible spatial variability of the Bering Strait current.**

7) The idea that the Nome River Glaciation started during peak warmth in MIS 11 is a bit counter-intuitive; I think a better treatment of the extant Nome River Glaciation sites (and in particular, their respective age controls) is required to support this idea. Also, while the authors do introduce the "snow-gun" hypothesis near the end of Subsection 5.3.2, I think re-organization to increase clarity and introduce the snow-gun idea sooner will greatly improve the readability here.

**We agree that it is counter-intuitive to find that the Nome River Glaciation began during peak warmth, however, we believe that there is significant terrestrial evidence that supports this not only occurred during MIS 11, but also during MIS 5e (Kaufman and Brigham-Grette, 1993; Kaufman et al., 2001; Pushkar et al., 1999). What these terrestrial studies lack is an accurate chronology. We think that our findings can add support to the onshore chronology and provide an important means of linking land and sea responses. We will rewrite this section to make this more clear. Additionally, we recognize that this hypothesis is not adequately tested yet but rather it provides an opportunity for future work.**

There are a few minor issues as well:

1) Overall, the mean d15N = 6.4‰ for the full dataset, and from looking at Fig. 7, it looks like there might be values that exceed 8 or 9‰These high values are suggestive of denitrification, yet this process isn't considered ˙ in the N cycle discussions spread throughout the paper.

**Yes, thank you for pointing out this omission. You are likely correct that denitrification is happening thoughout much of the record. This will be expanded upon and the new analyses you suggested above should clarify the isotope results.**

2) Because many of the figures are very data-rich, in many cases axes have been truncated, which makes it difficult to assess extreme data points (which are often very important, such as the extremely low d15N values associated with the 406-402 ka event). I would recommend that, instead of cutting axes ranges, they should instead be offset so that the full axis range can be indicated. I'm specifically thinking of Figure 7, but this could apply to many other figures, too. There are also several instances where it is difficult to determine which line goes with which axis; perhaps color coding or additional labels are necessary.

**We tried hard to make the figures as readable as possible. Thank you for these very specific suggestions to help us improve.**

3) I am also providing a PDF copy of the manuscript that I have made several grammatical corrections to; please review in detail.

In conclusion, I would like to recommend this article for acceptance, pending the minor revisions I've indicated here, as well as the editorial revisions on the attached manuscript. If any of my notes are not clear (or legible), I recommend the authors contact me directly with any questions they may have.

Please also note the supplement to this comment:

**Noted, thank you for the detailed comments.**

[revised manuscript text omitted]

Relative percent abundances of the characteristic marginal ice zone species, *F. oceanica* and *F. cylindrus* {Caissie, 2010 #900;von Quillfeldt, 2003 #677;Saito, 1978 #804;Sancetta, 1982 #213}, oscillate between ~10% and less than 3% of the diatom assemblage and are highest during MIS 12 and all laminated intervals. They are both at their lowest between ~411 to ~400 ka (126.62 to 123.45 mbsf). *L.* cf. *ocellata* is the dominant taxa in the fresh water group

| Page 12: [13] Deleted | Beth Caissie | 5/16/16 12:52 PM |
|---|---|---|

The neritic species and moving water indicator, *P. sulcata* is lowest during the laminated intervals. It reaches a maximum (34% relative abundance) at 404 ka (124.61 mbsf). *P. sulcata* remains moderately high (~10%) during non-laminated intervals. *L.* cf. *ocellata* is the dominant taxa in the fresh water group and the variability in its abundances is discussed below. *S. trifultus* follows a very similar distribution to the fresh water group and *L.* cf. *ocellata*. It is relatively high (~4%) during MIS 12, is virtually absent from the sediments during the Termination V Laminations, and then increases again until it peaks at 10% relative abundance at 400 ka (123.22 mbsf). *Thalassiosira binata* and other small (<10 μm in diameter) *Thalassiosira* species have similar distributions with low relative abundances throughout the record (< 6%) except for a small peak between 397 and 386 ka (122.62 and 119.07 mbsf.

| Page 12: [14] Formatted | Beth Caissie | 5/16/16 12:50 PM |
|---|---|---|

paragraph-chapters,  No bullets or numbering

[revised manuscript text omitted]

1.3.2

| Page 18: [45] Formatted | Beth Caissie | 6/16/16 9:57 AM |
|---|---|---|

paragraph-chapters, Justified, Space Before: 6 pt, Line spacing: 1.5 lines

| Page 23: [46] Formatted | Beth Caissie | 7/4/16 1:58 AM |
|---|---|---|

Not Highlight

| Page 23: [47] Formatted | Beth Caissie | 7/4/16 1:58 AM |
|---|---|---|

Not Highlight

| Page 23: [48] Moved from page 27 (Move #9)Beth Caissie | 6/16/16 11:26 AM |
|---|---|

Most of these laminations show an increase in sea ice diatoms and a decrease in productivity indicators. These roughly correspond with millennial scale stadial events that occurred during MIS 11a in the North Atlantic (Fig. 7) {Voelker, 2010 #4617}. Late

MIS 11 is characterized as a series of warm and cold cycles {Voelker, 2010 #4617;Candy, 2014 #4566}, though there is not agreement on the timing of these cycles.

[revised manuscript text omitted]

Normal, Left, Space Before: 0 pt, Line spacing: single, No bullets or numbering

| Page 23: [51] Moved from page 17 (Move #7)Beth Caissie | 6/16/16 10:03 AM |
|---|---|

At the same time, episodic increases in productivity were occurring in places as distant as Lake Baikal {Prokopenko, 2010 #1887} and the North Atlantic {Poli, 2010 #1892;Chaisson, 2002 #1890;Dickson, 2009 #1899} and NADW formation was intensifying {Poli, 2010 #1892}. Ventilation of NADW generally continues to increase from 424 to 410 ka to its strongest and then weakens over the course of the interglacial {Thunell, 2002 #1891}. In addition, the flux of terrigenous dust was decreasing near Antarctica reflecting perhaps a decrease in the strength of Southern Ocean winds {Wolff, 2006 #509}. Evidence for higher productivity in the Bering Sea, possibly caused by intensified upwelling, suggests teleconnections between NADW formation, the strength of the southern winds, upwelling in the North Pacific, and northward flow through Bering Strait {e.g. \DeBoer, 2004 #24}.

| Page 23: [52] Moved from page 18 (Move #8)Beth Caissie | 6/16/16 10:05 AM |
|---|---|

During MIS 11c, global ice volume was the lowest that it has been for the past 500 kyrs {Lisiecki, 2005 #1520}, and generally continental temperatures were warmer than today {Vogel, 2013 #4569;D'Anjou, 2013 #4620;Melles, 2012 #2158;Pol, 2011 #1880;Lyle,

#149;Prokopenko, 2010 #1887;de Vernal, 2008 #1908;Tzedakis, 2010 #1888;Raynaud, 2005 #101;Tarasov, 2011 #1897;Lozhkin, 2013 #1965} with a northward expansion of boreal forests in Beringia {Kleinen, 2014 #4563}. However, it was not warm uniformly world-wide. At U1345, the relative percent warm water species suggest that SSTs during MIS 11c were only slightly warmer than during MIS 12. Indeed, MIS 11 is not the warmest interglacial in most marine records {Candy, 2014 #4566}. This is especially evident in the Nordic Seas where MIS 11 SSTs were lower than Holocene values, although no IRD was deposited between 408 and 398 ka {Bauch, 2000 #156}.

However, MIS 11c was very humid in many places. In the Bering Sea, modeling studies estimate up to 50 mm more precipitation than today at 410 ka {Kleinen, 2014 #4563}. The most humid, least continental period recorded in the sediments at Lake Baikal occurs from 420-405 ka {Prokopenko, 2010 #1887}, and extremely high precipitation are recorded at Lake El'gygytgyn on the nearby Chukotka Peninsula from 420-400 ka {Melles, 2012 #2158}. Conditions in Africa during MIS 11c were similar to the Holocene African humid period. In addition, pollen records from Western Europe also reflect humid environments {Candy, 2014 #4566}. A warmer, moister climate in Western Europe and Africa is indicative of increased Atlantic Meridional Overturning Circulation (AMOC) {Bauch, 2013 #4619}. AMOC appears to be stable over MIS 11 {Milker, 2013 #4568} as evidenced by high carbonate in the North Atlantic {Poli, 2010 #1892;Chaisson, 2002 #1890}. Interestingly, small carbonate peaks in the Bering Sea are contemporaneous with those on the Bermuda Rise, suggesting teleconnections between the two regions (Fig. 7). These conditions are similar to a modern day negative North Atlantic Oscillation (NAO) which is linked to wet conditions in N. Africa, weaker westerlies, more zonal storm tracks, a dry Northern Europe, colder Nordic Seas and increased sea ice in the North Atlantic {Kandiano, 2012 #4570}.

| Page 23: [53] Formatted | Beth Caissie | 7/4/16 1:58 AM |
|---|---|---|

Font:(Default) +Theme Body, 11 pt, Font color: Auto, English (US)

| Page 23: [54] Formatted | Beth Caissie | 6/23/16 1:45 PM |
|---|---|---|

Formatted

| Page 23: [55] Formatted | Beth Caissie | 7/4/16 1:58 AM |
|---|---|---|

Font:Arial, Font color: Black, English (UK), Kern at 16 pt

| Page 23: [56] Formatted | Beth Caissie | 6/16/16 4:34 PM |

No bullets or numbering

| Page 23: [57] Deleted | Beth Caissie | 6/16/16 4:31 PM |

Between 405 and 394 ka, there is an unusual diatom assemblage and grain size distribution at Site U1345. There are several possible explanations for deposition of shallow water and fresh water species along with large changes in sediment grain size. We will consider two possibilities in detail: Bering Strait current reversal and glacial surge in Beringia.

| Page 23: [58] Deleted | Beth Caissie | 7/4/16 1:14 AM |

resulting from increased illite deposition

[revised manuscript text omitted]

{Voelker, 2010 #4617;Candy, 2014 #4566}{Voelker, 2010 #4617}{Chaisson, 2002 #1890}{Rohling, 2010 #1903}{Knudson, 2015 #4651}{Voelker, 2010 #4617}{EPICA community members, 2004 #38}{Vogel, 2013 #4569}

| Page 27: [64] Deleted | Beth Caissie | 7/3/16 2:39 PM |
| --- | --- | --- |

{Voelker, 2010 #4617;Candy, 2014 #4566}{Voelker, 2010 #4617}{Chaisson, 2002 #1890}{Rohling, 2010 #1903}{Knudson, 2015 #4651}{Voelker, 2010 #4617}{EPICA community members, 2004 #38}{Vogel, 2013 #4569}

| Page 27: [65] Formatted | Beth Caissie | 7/4/16 1:58 AM |
|---|---|---|
| Font:Times New Roman, 12 pt | | |

| Page 27: [65] Formatted | Beth Caissie | 7/4/16 1:58 AM |
|---|---|---|
| Font:Times New Roman, 12 pt | | |

| Page 27: [65] Formatted | Beth Caissie | 7/4/16 1:58 AM |
|---|---|---|
| Font:Times New Roman, 12 pt | | |

| Page 27: [65] Formatted | Beth Caissie | 7/4/16 1:58 AM |
|---|---|---|
| Font:Times New Roman, 12 pt | | |

| Page 27: [66] Deleted | Beth Caissie | 7/4/16 1:32 AM |
|---|---|---|
| glacial | | |

| Page 27: [66] Deleted | Beth Caissie | 7/4/16 1:32 AM |
|---|---|---|
| glacial | | |

| Page 27: [67] Deleted | Beth Caissie | 7/4/16 1:32 AM |
|---|---|---|
| Throughout much of MIS 11, | | |

| Page 27: [67] Deleted | Beth Caissie | 7/4/16 1:32 AM |
|---|---|---|
| Throughout much of MIS 11, | | |

| Page 27: [68] Formatted | Beth Caissie | 7/4/16 1:58 AM |
|---|---|---|
| Not Highlight | | |

| Page 27: [68] Formatted | Beth Caissie | 7/4/16 1:58 AM |
|---|---|---|
| Not Highlight | | |

| Page 27: [69] Deleted | Beth Caissie | 7/4/16 12:56 AM |
|---|---|---|

There is inconclusive evidence for a reversal of the Bering Strait current at 405 ka, but evidence for teleconnections between the Atlantic and the North Pacific is strong when eustatic sea level fluctuated near the Bering Strait sill depth at the end of MIS 11. Tidewater glaciers advanced in Beringia when eustatic sea level was high, insolation was declining in the Arctic, and other high latitude regions saw decreasing SSTs.

| Page 27: [69] Deleted | Beth Caissie | 7/4/16 12:56 AM |
|---|---|---|

There is inconclusive evidence for a reversal of the Bering Strait current at 405 ka, but evidence for teleconnections between the Atlantic and the North Pacific is strong when eustatic sea level fluctuated near the Bering Strait sill depth at the end of MIS 11. Tidewater glaciers advanced in Beringia when eustatic sea level was high, insolation was declining in the Arctic, and other high latitude regions saw decreasing SSTs.

| Page 27: [69] Deleted | Beth Caissie | 7/4/16 12:56 AM |

There is inconclusive evidence for a reversal of the Bering Strait current at 405 ka, but evidence for teleconnections between the Atlantic and the North Pacific is strong when eustatic sea level fluctuated near the Bering Strait sill depth at the end of MIS 11. Tidewater glaciers advanced in Beringia when eustatic sea level was high, insolation was declining in the Arctic, and other high latitude regions saw decreasing SSTs.

| Page 27: [69] Deleted | Beth Caissie | 7/4/16 12:56 AM |

| Page 27: [70] Deleted | Beth Caissie | 7/4/16 12:16 AM |

During MIS 12, seasonal sea ice dominated the western Bering Sea with highly stratified waters during the ice-melt season. Sea ice was at a minimum from 423 to 410 ka when the Termination V Laminations were deposited. After this, although summer

| Page 27: [70] Deleted | Beth Caissie | 7/4/16 12:16 AM |

| Page 27: [70] Deleted | Beth Caissie | 7/4/16 12:16 AM |

| Page 27: [70] Deleted | Beth Caissie | 7/4/16 12:16 AM |

During MIS 12, seasonal sea ice dominated the western Bering Sea with highly stratified waters during the ice-melt season. Sea ice was at a minimum from 423 to 410 ka when the Termination V Laminations were deposited. After this, although summer

| Page 28: [71] Deleted | Beth Caissie | 7/4/16 1:02 AM |

Evidence of glaciation is short lived in the western Bering Sea and followed by an intensification of northerly winds that brought freshwater diatoms out over the open ocean.

| Page 28: [72] Formatted | Beth Caissie | 7/4/16 1:03 AM |

paragraph-chapters, Justified, Space Before:  6 pt, Line spacing:  1.5 lines

| Page 29: [73] Deleted | Beth Caissie | 7/4/16 12:59 AM |

Laminations at end MIS 11 correspond with millennial scale stadials seen in the N Atlantic. These deposits represent further possible evidence of teleconnections between the Atlantic and the Pacific as eustatic sea level fluctuated near the Bering Strait sill depth.

This study supports hypotheses that the region responds to insolation changes at 65° N and that Bering Strait modulates climate in both the North Atlantic and Pacific regions. Future work should focus on leads and lags between changes in the North Atlantic, North Pacific and Antarctic regions to determine how upwelling, deep water formation, and climate are related.

| Page 30: [74] Deleted | Beth Caissie | 7/3/16 2:29 PM |

---

## Editor Decision (ED1)

[revised manuscript text omitted]

---

## Author Response (AR2)

Thank you very much for pointing out several issues with the most recent version of our manuscript. We noticed a few others as well in our final read through. Below, we've included a summary of the changes made to this version of the manuscript.

Line 31: Replace "sync" with "synchrony." Replaced "brief" with "millennial-scale."
Line 32: Replaced "an" with "a."
Line 88: Added "history."
Lines 258-261: updated dates in Mineralogy section.
Lines 282 -284: Shortened genus names for genera already mentioned.
Line 289: Removed extra paragraph marker and typos.
Line 290: Replaced "and these optima" with "that."
Lines 328-329: updated epontic diatoms results after transfer of several species out of this niche.
Lines 510-511: changed description of sea ice reduction to reflect new information from epontic species.
Line 617: Changed "tantalizing" to "possible"
Line 700: updated mineralogy date.
Line 843: included data archive web address.

Table 3: Species names updated in several places.

The following references that only appear in figure captions were added to the list of references: Barr and Clark, 2009; Cavalieri, et al., 1996; Gualtieri, et al., 2000; Heiser and Roush, 2001; Kaufman et al., 2011; Kim et al., 2015; Manley, 2002; Meyers, 1994; Redfield, 1963; and Walinsky, et al., 2009.

Figure 7: Changed cells/g to calcareous nannofossils/gram in figure and in caption. Revised epontic diatom plot in light of change in species niche in Table 3.

[revised manuscript text omitted]